# Fix the Loss, Not the Radius:
# Rethinking the Adversarial Perturbation of Sharpness-Aware Minimization

Jinping Wang [1 2]   Qinhan Liu [2]   Zhiwu Xie [2]   Zhiqiang Gao [* 1 2]

## Abstract

Sharpness-Aware Minimization (SAM) improves generalization by minimizing the worst-case loss within a fixed parameter-space radius neighborhood. SAM and its variants mainly rely on a first-order linearized surrogate, while flat minima are inherently a second-order (curvature) notion. We revisit this mismatch and propose Loss-Equated SAM (LE-SAM), which inverts the traditional SAM mechanism that replaces the fixed perturbation radius with a fixed loss-space budget, effectively removing gradient-norm–dominated learning signals and shifting optimization toward curvature-dominated terms. Extensive experiments across diverse benchmarks and tasks demonstrate the strong generalization ability of LE-SAM that consistently outperforms SAM and even its variants, achieving state-of-the-art performance. https://github.com/smithgun2005/LE-SAM-ICML2026

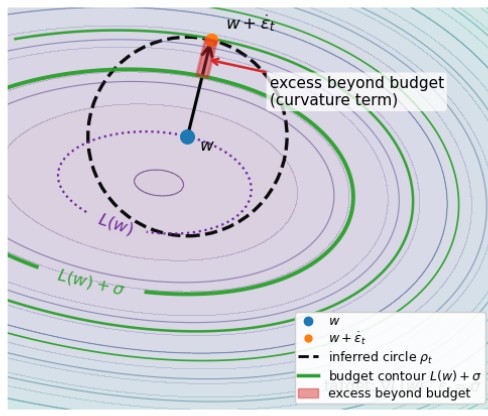

*Figure 1.* Illustration of loss-equated adversarial perturbation. LE-SAM fixes a budget in loss space and back-solves the corresponding adversarial perturbation radius in parameter space. This mechanism removes gradient norm dominated effects and shifts the optimization signal toward curvature-dominated (second-order) information. The shaded region indicates the excess loss beyond the budget due to the curvature term.

## 1. Introduction

Deep Neural Networks (DNNs) have been used for various tasks and achieved remarkable performance (Devlin et al., 2019; Brown et al., 2020; Liu et al., 2021). However, due to the large number of model parameters, modern DNNs are often highly over-parametrized with highly non-convex loss landscapes, leading to various local optimal points. Recent studies (Keskar et al., 2017; Dziugaite & Roy, 2017; Jiang et al., 2020) have suggested that different local optimal points may have different generalization abilities. Flat minima, characterized by uniformly low loss values are shown to always lead to better generalization performance, while

---
*Corresponding author. [1]International Frontier Interdisciplinary Research Institute, Wenzhou-Kean University [2]CSMT, Wenzhou-Kean University. Correspondence to: Zhiqiang Gao <zgao@wku.edu.cn>, Jinping Wang <1306325@wku.edu.cn>, Qinhan Liu <1306303@wku.edu.cn>, Zhiwu Xie <zxie@wku.edu.cn>.

*Proceedings of the 43rd International Conference on Machine Learning*, Seoul, South Korea. PMLR 306, 2026. Copyright 2026 by the author(s).

sharp minima might cause abrupt loss changes and lead to unstable generalization performance.

To seek flat minima, Sharpness-Aware Minimization (SAM) (Foret et al., 2021) is proposed, which optimizes a worst-case loss proxy via an adversarial ascent perturbation step. Specifically, SAM leverages the first-order linear approximation to seek the neighborhood worst point with the highest loss within the fixed perturbation radius $\rho$. This adversarial mechanism enables the model to seek local optima characterized by consistently low-loss neighborhoods, resulting in better generalization performances.

Despite the empirical success of SAM and its variants (Kwon et al., 2021; Du et al., 2022; Zhuang et al., 2022; Ji et al., 2024; Li et al., 2024) in improving the generalization of modern DNNs, we argue that the learning objective of current methods is mismatched with the ideal objective for achieving the flat minima. In particular, the concept of flatness is inherently a second-order property of the loss

landscape characterized by the local curvature around a minimum. However, SAM constructs its adversarial objective via a first-order linear approximation, which mainly *penalizes the gradient norm* within a fixed perturbation radius $\rho$ instead of directly penalizing the curvature of the loss surface (Wu et al., 2024; Luo et al., 2025).

This observation inspires us to rethink and systematically analyze the adversarial mechanism of SAM. Our investigation reveals a critical discrepancy. While the ideal concept of flat minima is fundamentally dominated by second-order properties, such as the local curvature around a minimum, the practical objective of SAM is actually heavily dominated by first-order gradient magnitudes rather than the second-order curvature it aims to optimize. This gradient-dominated learning signal distracts the optimizer from directly penalizing the loss surface sharpness, creating a fundamental discrepancy between the training mechanism and the generalization goal. Consequently, SAM often becomes effective only in the late phase of training when gradients are small enough for curvature signals to finally become comparable (Zhou et al., 2025).

This insight motivates us to further unleash the full potential of second-order information by redesigning the adversarial perturbation mechanism of SAM. Distinct from existing curvature-aware SAM variants that often append second-order metrics to the original framework while retaining the first-order perturbation, for example, adding a Hessian-trace regularizer (Wu et al., 2024), explicitly targeting the top Hessian eigenvalue via eigenvector alignment (Luo et al., 2025) or modifying the neighborhood geometry via a Fisher-information based metric (Kim et al., 2022), our approach is a mechanism-level modification that completely abandons the first-order gradient signal in the adversarial step. This allows the curvature-based signals to become the dominant learning signal throughout the entire training process and provides a more direct and robust path to flat regions. Furthermore, as previous work points out that the first-order term or the gradient norm is highly unstable due to the drastic variance of gradient magnitudes across mini-batches (Tan et al., 2024), our method explicitly removing this signal, which further enhances optimization stability compared to traditional gradient-norm-based approaches.

To realize this refined optimization goal, we propose Loss-Equated SAM (LE-SAM), a reformulation that inverts the traditional mechanism by fixing the loss instead of the radius. Inspired by the classical Polyak step-size (Polyak, 1987), we scalarize the first-order loss variation into a fixed loss-space budget $\sigma$. This transformation ensures the first-order term becomes a constant that is independent of the model parameters, which effectively removes its interference and shifts the learning signal toward curvature-dominated terms. Under this framework, we derive a closed-form dynamic perturba-

tion radius $\rho_t$ that adaptively adjusts to the local landscape. As illustrated in Figure 1, this dynamic radius allows the model to consistently reach the budget contour and focus its update on the excess beyond budget produced by the pure curvature term, thereby identifying the flattest minima with higher precision throughout the entire training process. As such, our methods offer a unified view of sharpness-aware learning that is intrinsically curvature-aware, leading to the following main contributions:

- We identify a mechanism-level objective mismatch in Sharpness-Aware Minimization (SAM): the adversarial perturbation is generated by solving the inner maximization via a fixed-radius constraint using a first-order linearized surrogate, making the learning signal largely dominated by gradient magnitude, whereas the ideal notion of flat minima is inherently a second-order concept.

- We propose LE-SAM, a mechanism-level reformulation of SAM that mitigates the mismatch by redesigning the adversarial perturbation to a loss-equated framework. Through fixing the perturbation loss budgets in loss space, LE-SAM scalarizes the first-order variation and shifts the optimization signal towards the second-order curvature terms.

- Extensive experiments demonstrate that through such a mechanism-level redesign of SAM, our LE-SAM can consistently outperform SAM and existing SAM variants, achieving the state-of-the-art (SOTA) performance.

## 2. Notation and Problem Formulation

We draw an i.i.d. training dataset $\mathcal{S} \triangleq \bigcup_{i=1}^{n} \{(\mathbf{x}_i, \mathbf{y}_i)\}$ from the population distribution $\mathcal{D}$. Our goal is to train a model using only the training dataset $\mathcal{S}$ that can generalize well on the population distribution $\mathcal{D}$. Consider a family of models parameterized by $\mathbf{w} \in \mathcal{W} \subseteq \mathbb{R}^d$; For the loss function $l : \mathcal{W} \times \mathcal{X} \times \mathcal{Y} \to \mathbb{R}_+$, we denote $L_{\mathcal{S}}(\mathbf{w}) \triangleq \frac{1}{n} \sum_{i=1}^{n} l(\mathbf{w}, \mathbf{x}_i, \mathbf{y}_i)$ as the loss on the training set and $L_{\mathcal{D}}(\mathbf{w}) \triangleq \mathbb{E}_{(x,y)\sim\mathcal{D}} [l(\mathbf{w}, \mathbf{x}, \mathbf{y})]$ as the loss for the population.

## 3. Revisiting Sharpness-Aware Minimization

In this section, we revisit SAM from a mechanism-level perspective. We point out the widely adopted first-order adversarial perturbation often makes the optimization signal largely gradient norm-driven, creating a mechanism-level mismatch with the ideal target of flat minima.

### 3.1. Learning Objective for SAM

Deep Neural Networks are heavily over-parameterized, the loss landscape of a DNN is often highly non-convex and complex, leading to numerous local optimal points. However, different local optimal points may lead to a huge difference in the generalization ability of the model. A widely accepted idea is that the flat minima can have a stronger generalization performance. To seek flat minima, (Foret et al., 2021) proposed Sharpness-Aware Minimization (SAM), a simple and effective algorithm that operationalizes this intuition by minimizing the worst-case loss in the neighborhood. The optimization problem of SAM can be described as follows:

$$\min_{\mathbf{w}} \ L_{\mathcal{S}}^{\mathrm{SAM}}(\mathbf{w}) + \lambda \|\mathbf{w}\|_2^2 \quad,$$
$$\text{where} \quad L_{\mathcal{S}}^{\mathrm{SAM}}(\mathbf{w}) \triangleq \max_{\|\boldsymbol{\epsilon}\|_2 \leq \rho} L_{\mathcal{S}}(\mathbf{w} + \boldsymbol{\epsilon}). \quad (1)$$

Here, $\epsilon$ denotes an adversarial perturbation in the parameter space, $\rho > 0$ is the perturbation radius that controls the size of the adversarial neighborhood, and $\lambda \|\mathbf{w}\|_2^2$ denotes the standard weight decay. Solutions that maintain low loss across the parameter neighborhood are more robust to parameter perturbations and thus are biased toward flatter regions of the loss landscape.

### 3.2. Inner Maximization and Adversarial Perturbation for SAM

SAM is driven by an explicit two-step generate–evaluate–update mechanism: starting from the current parameters $\mathbf{w}$, it first constructs an adversarial perturbation $\boldsymbol{\epsilon}$ that seeks the adversarial point that has the highest loss within a local neighborhood, and then updates $\mathbf{w}$ using the gradient evaluated at the adversarially perturbed point.

Since the inner maximization $L_{\mathcal{S}}^{\mathrm{SAM}}(\mathbf{w}) \triangleq \max_{\|\boldsymbol{\epsilon}\|_2 \leq \rho} L_{\mathcal{S}}(\mathbf{w} + \boldsymbol{\epsilon})$ for SAM is hard to compute in practice. To approximate the worst-case loss in the neighborhood, SAM uses the following first-order Taylor approximation :

$$L_N(\mathbf{w}) = L_{\mathcal{S}}(\mathbf{w} + \hat{\epsilon}) \quad \text{where} \ \hat{\epsilon} = \rho \frac{\nabla_{\mathbf{w}} L_{\mathcal{S}}(\mathbf{w})}{\|\nabla_{\mathbf{w}} L_{\mathcal{S}}(\mathbf{w})\|_2} \quad (2)$$

Here, $\hat{\epsilon}$ is the approximated adversarial perturbation and $L_N(\mathbf{w})$ is the approximated worst-case loss in the neighborhood. Then SAM computes the gradient at these perturbed parameters:

$$g_{SAM} = \nabla_{\mathbf{w}} L_N(\mathbf{w}), \quad (3)$$

and use $g_{SAM}$ to update the model parameters $\mathbf{w}$.

Notably, $\hat{\epsilon}$ is obtained from a first-order linearization and is aligned with the gradient direction. The practical SAM surrogate is often primarily shaped by the first-order gradient signal. In contrast, flat minima are fundamentally a second-order concept. The next section formalizes this mechanism-level mismatch.

### 3.3. Mismatch with Flat Minima

For a local minima at $\mathbf{w}^*$, consider the second-order Taylor expansion takes the form:

$$L(\mathbf{w}^* + \epsilon) = L(\mathbf{w}^*) + \underbrace{\nabla L(\mathbf{w}^*)^{\top}\epsilon}_{\approx 0} + \frac{1}{2}\epsilon^{\top} H(\mathbf{w}^*)\epsilon + \mathcal{O}(\|\epsilon\|^3),$$
$$(4)$$

Here, $H(\mathbf{w}^*)$ is the Hessian. Near the local minima, the gradient is approximately 0 ($\nabla L(\mathbf{w}^*) \approx 0$), thus the leading term that governs how fast the loss increases around $\mathbf{w}^*$ is:

$$L(\mathbf{w}^* + \epsilon) - L(\mathbf{w}^*) \approx \frac{1}{2}\epsilon^{\top} H(\mathbf{w}^*)\epsilon \quad (5)$$

Thus, flat minima is inherently a second-order concept: *the flatness of a minima is often characterized by small Hessian eigenvalues*. Based on this, many approaches adopt the Hessian-based notion of flatness and explicitly characterize flatness via the spectrum of the Hessian (Wu et al., 2024; Luo et al., 2025; Foret et al., 2021) to modify SAM.

Now, recall the update mechanism for SAM we discussed in the previous section, we can expand the Eq. 2:

$$L_N(\mathbf{w}) = L_{\mathcal{S}}(\mathbf{w} + \hat{\epsilon})$$
$$= L_{\mathcal{S}}(\mathbf{w}) + \nabla L_{\mathcal{S}}(\mathbf{w})^{\top}\hat{\epsilon} + \frac{1}{2}\hat{\epsilon}^{\top} H(\mathbf{w})\hat{\epsilon} + \mathcal{O}(\|\hat{\epsilon}\|^3)$$
$$= L_{\mathcal{S}}(\mathbf{w}) + \rho\|\nabla L_{\mathcal{S}}(\mathbf{w})\|_2 + \frac{1}{2}\hat{\epsilon}^{\top} H(\mathbf{w})\hat{\epsilon} + \mathcal{O}(\|\hat{\epsilon}\|^3).$$
$$(6)$$

From the above expansion, we can observe that the practical SAM surrogate $L_N(\mathbf{w})$ approximates the local sharpness *along a single gradient aligned*. Most of the training process is dominated by the first-order gradient norm $\rho\|\nabla L_S(\mathbf{w})\|$. Thus, *only in the late phase of training, when gradients are small, the curvature term $\frac{1}{2}\hat{\epsilon}^{\top} H(\boldsymbol{w})\hat{\epsilon}$ become comparable*, which matches the observation by (Zhou et al., 2025) that SAM is generally effective only at the later stage of training.

Thus, this mismatch is an intrinsic consequence of the adversarial perturbation mechanism of SAM. Mitigating the mismatch, therefore, requires redesigning how the inner adversarial perturbation is constructed.

### 3.4. SAM Variants

In recent years, many SAM variants have been proposed by modifying different parts of the algorithm. The modification for these SAM variants can be concluded mainly in

three aspects, targeting improved generalization, stability, or efficiency.

The first group of variants focuses on changing and improving the metric in parameter space to construct more effective perturbation. For example, ASAM (Kwon et al., 2021) scales the the gradient to obtain scale-invariant perturbations. Fisher-SAM (Kim et al., 2022) uses a Fisher-like metric to shape the ascent perturbation. Eigen-SAM (Luo et al., 2025) estimates the top Hessian eigenvector and injects such information into SAM perturbation.

The second group of work modifies how the gradient at the adversarial point is used, or how the perturbation loss is combined with the clean loss or other regularization terms. For example, GSAM (Zhuang et al., 2022) decomposes the SAM gradient into two component and discarding the conflict component with respect to the vanilla descent direction. Friendly SAM (Li et al., 2024) also decomposes the perturbation direction of SAM into a full-gradient direction and a stochastic noise direction, showing that the noise component is the main driver of generalization.

The third group of work focuses on the stability and the efficiency of SAM. By sharing computations and subsampling data , ESAM (Du et al., 2022) make SAM more efficient. SSAM (Tan et al., 2024) rescales the gradient norm at the perturbed point to match the center gradient norm to improve the stability of SAM.

In short, better metrics, smarter gradient combinations or other options can offer a certain level of performance improvement over SAM, leading to better generalization performance. However, at the mechanism level, most still retain the same core: **generating adversarial perturbations under a fixed radius $\rho$ using a first-order linearized surrogate.** As a result, curvature information still tends to enter in a highly compressed manner, and the objective mismatch highlighted in Section 3.3 is not fundamentally resolved. This motivates our mechanism-level redesign in the next section: **fix the loss, not the radius.**

## 4. Proposed Method

The previous section demonstrates the objective mismatch of the SAM and SAM variants. To address such a mismatch, inspired by the traditional optimization approach Polyak step-size (Polyak, 1987), we propose our Loss-Equated SAM (LE-SAM) algorithm, which inverts the traditional perturbation design of the ascent step for SAM.

### 4.1. Polyak Inspiration

The inspiration of our algorithm is the traditional Polyak step-size (Polyak, 1987), an adaptive step-size strategy that fixes an ideal target decrease in the objective and derives the

---

**Algorithm 1** Loss-Equated Sharpness-Aware Minimization (LE-SAM) with Clipping & Annealing

---

**Require:** Training data $\mathcal{D}$, loss $\ell(\mathbf{w}; \mathbf{x}, \mathbf{y})$, base optimizer OPT (e.g., SGD/Adam), initial loss budget $\sigma_0 > 0$, stability constant $\varrho > 0$, radius cap $\rho_{\max} > 0$, annealing length $n$ (epochs) and total epochs $T$ (or total steps $K$)

1: Initialize parameters $\mathbf{w}$
2: **for** each training step $t = 1, 2, \dots$ **do**
3:      Sample a mini-batch $\mathcal{B} = \{(\mathbf{x}_i, \mathbf{y}_i)\}_{i=1}^m$ from $\mathcal{D}$
4:      $\mathcal{L}(\mathbf{w}) \leftarrow \frac{1}{m} \sum_{(\mathbf{x},\mathbf{y})\in\mathcal{B}} \ell(\mathbf{w}; \mathbf{x}, \mathbf{y})$
5:      $\mathbf{g} \leftarrow \nabla_{\mathbf{w}} \mathcal{L}(\mathbf{w})$ {center gradient}
6:      (Anneal loss budget)
     $\sigma_t \leftarrow \text{Cosine Anneal}(\sigma_0, t; n, T)$ {$\sigma_t \to 0$ over last $n$ epochs}
7:      $\rho_t \leftarrow \frac{\sigma_t}{\|\mathbf{g}\|_2 + \varrho}$ {loss-equated radius}
8:      **(Clip radius)** $\rho_t \leftarrow \min(\rho_t, \rho_{\max})$ {radius clipping for stability}
9:      $\boldsymbol{\epsilon}_t \leftarrow \rho_t \frac{\mathbf{g}}{\|\mathbf{g}\|_2}$ {adversarial perturbation}
10:     $\widehat{\mathcal{L}}(\mathbf{w}) \leftarrow \frac{1}{m} \sum_{(\mathbf{x},\mathbf{y})\in\mathcal{B}} \ell(\mathbf{w} + \boldsymbol{\epsilon}_t; \mathbf{x}, \mathbf{y})$
11:     $\widehat{\mathbf{g}} \leftarrow \nabla_{\mathbf{w}} \widehat{\mathcal{L}}(\mathbf{w})$ {sharpness-aware gradient}
12:     $\mathbf{w} \leftarrow \text{OPT}(\mathbf{w}, \widehat{\mathbf{g}})$ {update with base optimizer}
13: **end for**
14: Return $\mathbf{w}$

---

corresponding descent magnitude.

Consider the optimization problem

$$\min_{x \in \mathbb{R}^n} f(x), \tag{7}$$

where $f(x)$ is a convex function and $f^*$ denotes its optimal value. Let $g_k$ be a (sub)gradient of $f$ at iteration $k$. The Polyak step-size is defined as

$$\alpha_k = \frac{f(x_k) - f^*}{\|g_k\|^2}. \tag{8}$$

The update rule is then given by

$$x_{k+1} = x_k - \alpha_k g_k. \tag{9}$$

Intuitively, this step-size adjusts the magnitude of the update according to the distance from optimality. By fixing the target descent, descent is no longer dominated by gradient magnitude, yielding a theoretically grounded mechanism, leading to faster convergence and reduced oscillations. Motivated by this, we try to transfer this idea to the adversarial mechanism of SAM, which will be discussed in the following section.

### 4.2. Loss-Equated Perturbation

Recall that for the standard SAM ascent step in Eq. 2, denote $g_c$ as the gradient of the original model weights where $g_c =$

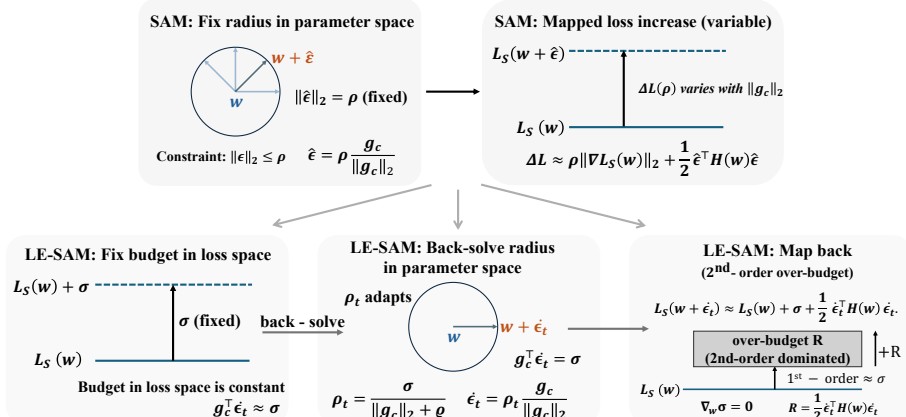

*Figure 2.* Mechanism illustration of SAM vs. LE-SAM. Standard SAM fixes a perturbation radius in the parameter space and directly maps it to a loss increase dominated by the gradient norm. In contrast, LE-SAM fixes a budget in the loss space and back-solves the perturbation radius in the parameter space. Mapping back to the loss space removes the first-order gradient-dominated term and yields a second-order, curvature-dominated over-budget region. This mechanism-level redesign shifts the sharpness-aware objective from gradient magnitude dominated to more curvature information, matching the notion of flat minima.

$\nabla_{\mathbf{w}} L_{\mathcal{S}}(\mathbf{w})$. The standard SAM fixes a parameter-space radius $\rho$ and maximizes the linear term $g_c^\top \epsilon$ under $\|\epsilon\|_2 \leq \rho$ with the maximizer $\hat{\epsilon} = \rho \frac{g_c}{\|g_c\|_2}$. The first-order Taylor approximation gives:

$$L_{\mathcal{S}}^{SAM}(\mathbf{w}) - L_{\mathcal{S}}(\mathbf{w}) \approx L_N(\mathbf{w}) - L_{\mathcal{S}}(\mathbf{w}) \approx g_c^\top \hat{\epsilon} = \rho \|g_c\|_2 \tag{10}$$

Thus, the loss increase is actually changing whenever the gradient norm changes, resulting in unstable training and compressed second-order information.

Inspired by the classical Polyak step-size (Polyak, 1987), which fixes the reduced loss to calculate an optimal descent step-size, in LE-SAM, we invert this relationship. At time step t, instead of fixing $\rho$ like traditional SAM and its variants, we fix a loss budget $\sigma > 0$ in the loss space and choose the perturbation $\dot{\epsilon}_t$ to let the first-order loss match this budget:

$$g_c^\top \dot{\epsilon}_t \approx \sigma \tag{11}$$

Thus, we can get the radius $\rho_t$ and the perturbation $\dot{\epsilon}_t$ at time step t:

$$\dot{\epsilon}_t = \rho_t \frac{g_c}{\|g_c\|_2}, \quad \rho_t = \frac{\sigma}{\|g_c\|_2 + \varrho} \tag{12}$$

where $\varrho > 0$ is a small constant for numerical stability. Then we substitute $\dot{\epsilon}_t$ to obtain our adversarial loss $\hat{L}_N$ where:

$$\hat{L}_N(\mathbf{w}) = L_{\mathcal{S}}(\mathbf{w} + \dot{\epsilon}_t) \approx L_{\mathcal{S}}(\mathbf{w}) + \underbrace{\sigma}_{\text{scalar constant}} + \frac{1}{2} \dot{\epsilon}_t^\top H(\mathbf{w}) \dot{\epsilon}_t. \tag{13}$$

Eq. 13 illustrates the core mechanism of our algorithm: the first-order gradient norm term becomes a scalar constant $\sigma$ that does not depend on $\mathbf{w}$ in this surrogate. Following the standard SAM algorithm, we treat the inner perturbation $\epsilon_t$

as a constant in the outer minimization (i.e., stop-gradient through $\epsilon_t$), and only differentiate $L(w + \epsilon_t)$ with respect to $w$. Under this convention, the scalar term $\sigma$ has zero gradient where:

$$\nabla_{\mathbf{w}} \sigma = 0 \tag{14}$$

In contrast to standard SAM and its variants, where the surrogate loss contains a gradient-norm penalty $\nabla L_S(\mathbf{w})$, LE-SAM removes this gradient-norm penalty from the surrogate and replaces it with a fixed loss-space budget $\sigma$. The sharpness effect is no longer driven by *how large the gradient norm happens to be*, but by the second-order term $\frac{1}{2} \dot{\epsilon}_t^\top H(\mathbf{w}) \dot{\epsilon}_t$ evaluated at a perturbation whose magnitude in loss space is calibrated. This directly reflects the second-order notion of the flatness discussed in the previous section.

### 4.3. Practical Stabilization For Convergence

Since LE-SAM back-solves the perturbation radius, the radius can sometimes become excessively large when the gradient norm is small, leading to overly aggressive perturbations and oscillatory training. To improve the stability and ensure convergence, we adopt two simple safeguards. First strategy is **radius clipping**, we set a maximum value for the perturbation radius $\rho_{max}$ and let $\rho_t \leftarrow \min(\rho_t, \rho_{max})$ to prevent extreme perturbations. Another strategy we adopt is the **loss-budget annealing**, where we pre-anneal the loss budget $\sigma$ to 0 during the final stage of training (cosine decay over the last $n$ epochs), ensuring the perturbation smoothly vanishes, and the sharpness pressure is gradually turned off for stable final convergence. The algorithm of LE-SAM is shown in Algorithm 1.

# 5. Experiment

To empirically verify the effectiveness of our proposed methods, we conduct extensive experiments on different tasks and compare the performance with various SAM-based algorithms. We train our models using the Py-Torch toolbox (Paszke et al., 2019) on GeForce RTX4090 GPUs; We apply all methods with the same data augmentations following (Du et al., 2022) and evaluate the performance with three random seeds. Hyper-parameters for all the methods were either tuned to their optimal values or set according to the recommendations in the original paper. Further experiment details are shown in Appendix F.

**Baseline Methods** Since the field of Sharpness-Aware Minimization is developing rapidly, besides the traditional SAM (Foret et al., 2021), we choose ASAM (Kwon et al., 2021), ESAM (Du et al., 2022), F-SAM (Li et al., 2024), and Eigen-SAM (Luo et al., 2025) as baseline methods, which are either representative or recently published SAM variants with high generalization performance.

## 5.1. Training From Scratch

**CIFAR** We evaluate our methods over CIFAR10/100 (Krizhevsky et al., 2009) datasets, following the evaluation protocol of SAM and its variants (Du et al., 2022; Li et al., 2024; Foret et al., 2021; Kwon et al., 2021; Luo et al., 2025), we chose ResNet18 (He et al., 2016), WideResNet-28-10 (Zagoruyko & Komodakis, 2016), and PyramidNet-110 (Han et al., 2017) as experiment backbones. The experiment results are shown in Table 1. Notably, on CIFAR-100, our LE-SAM can outperform SAM by 1.34, 1.49, and 1.45 points while achieving the highest performance among all the SAM variants, demonstrating our loss-equated perturbation mechanism can effectively improve the generalization performance of the model.

**ImageNet** To evaluate the effectiveness of our method on large-scale dataset, we apply our LE-SAM on ImageNet (Deng et al., 2009) by training ResNet-50 and ResNet-101 from scratch. The experiment results are shown in Table 2. Our LE-SAM can outperform SAM by 0.71 and 0.77 points on two backbones, and consistently outperforms other SAM variants.

## 5.2. Fine-tuning Pre-trained Models (Transfer Learning)

We further evaluate our algorithm under the transfer learning scenario. We fine-tune a ViT-B-16 model (Dosovitskiy et al., 2021) pretrained on ImageNet and apply it on the CIFAR-10 and CIFAR-100 datasets. We use the pre-trained checkpoint provided by the official Py-Torch repository; all the methods are trained for 8k steps. Table 3 shows the

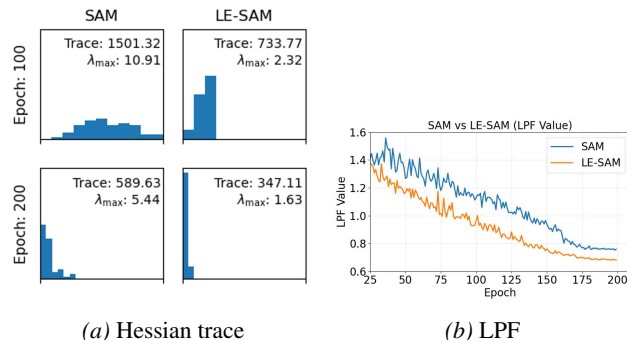

*Figure 3.* (a) shows the distribution of top eigenvalues and the trace of Hessian at epoch 100 and 200 on CIFAR-100 with SAM, and our LE-SAM. (b) tracks the LPF value for both SAM and LE-SAM across 200 epochs

experiment results, our LE-SAM consistently outperforms SAM on CIFAR datasets, and empirically demonstrate our loss-equated mechanism can lead to a better generalization performance in a transfer learning scenario.

## 5.3. Flatness Metrics

Eq. 13 shows that our loss-equated perturbation mechanism makes the first-order gradient-norm term a scalar that has no gradient with respect to the model parameters, thereby removing its dominance in the outer minimization and shifting the effective learning signal toward the curvature-dominated (second-order) terms. To empirically verify that our algorithm can effectively bias optimization toward flatter minima, we analyze the top eigenvalues and the trace of the Hessian matrix and track the LPF value (Bisla et al., 2022) during training.

We compute the Hessian spectra of ResNet-18 trained on CIFAR-100 for 200 epochs with SGD, SAM, and our LE-SAM algorithm. We apply power iteration (Yao et al., 2018) to compute the top eigenvalues of Hessian and Hutchinson method (Avron & Toledo, 2011; Bai et al., 1996; Yao et al., 2020) to compute the Hessian trace. Top-50 Hessian eigenvalues for each method are reported. As shown in Figure 3 (a), the model trained with LE-SAM has the lowest maximum Hessian eigenvalue and Hessian trace, moreover, shown in Figure 3 (b) our LE-SAM has lower LPF value during training. These observations match our analysis in the previous section, showing our method can effectively lead to a flatter minima. We show visualizations of landscapes of SGD, SAM, and LE-SAM in the following section.

## 5.4. Visualization of Landscapes

We visualize the loss landscapes (Li et al., 2018) of models trained with SGD, SAM, and our LE-SAM of the ResNet-18 model on CIFAR-100. All the models are trained with the optimal hyper-parameters. As shown in figure 4, we can

*Table 1.* Classification accuracies on the CIFAR-10 and CIFAR-100 datasets.

| CIFAR-10 | SGD | Eigen-SAM | ASAM | ESAM | F-SAM | SAM | LE-SAM(Ours) |
|---|---|---|---|---|---|---|---|
| ResNet-18 | $95.31_{\pm 0.07}$ | $96.50_{\pm 0.07}$ | $96.53_{\pm 0.12}$ | $96.56_{\pm 0.08}$ | $96.55_{\pm 0.08}$ | $96.52_{\pm 0.13}$ | $96.60_{\pm 0.11}$ |
| WRN-28-10 | $96.36_{\pm 0.09}$ | $97.36_{\pm 0.10}$ | $97.31_{\pm 0.17}$ | $97.29_{\pm 0.11}$ | $97.33_{\pm 0.12}$ | $97.27_{\pm 0.11}$ | $97.44_{\pm 0.06}$ |
| PyramidNet-110 | $96.56_{\pm 0.14}$ | $97.44_{\pm 0.09}$ | $97.46_{\pm 0.07}$ | $97.46_{\pm 0.01}$ | $97.46_{\pm 0.11}$ | $97.30_{\pm 0.10}$ | $97.63_{\pm 0.10}$ |
| CIFAR-100 | SGD | Eigen-SAM | ASAM | ESAM | F-SAM | SAM | LE-SAM(Ours) |
| ResNet-18 | $78.31_{\pm 0.10}$ | $80.52_{\pm 0.17}$ | $80.68_{\pm 0.17}$ | $80.71_{\pm 0.22}$ | $80.73_{\pm 0.21}$ | $80.17_{\pm 0.08}$ | $81.51_{\pm 0.08}$ |
| WRN-28-10 | $81.55_{\pm 0.14}$ | $83.54_{\pm 0.12}$ | $83.63_{\pm 0.20}$ | $83.88_{\pm 0.14}$ | $83.81_{\pm 0.21}$ | $83.42_{\pm 0.04}$ | $84.91_{\pm 0.11}$ |
| PyramidNet-110 | $81.88_{\pm 0.17}$ | $84.53_{\pm 0.09}$ | $84.50_{\pm 0.10}$ | $85.46_{\pm 0.08}$ | $85.48_{\pm 0.10}$ | $84.46_{\pm 0.04}$ | $85.91_{\pm 0.06}$ |

*Table 2.* Classification accuracies (%) on ImageNet

| ImageNet | SGD | Eigen-SAM | ASAM | ESAM |
|---|---|---|---|---|
| ResNet-50 | 76.02 | 76.94 | 76.68 | 77.02 |

| | F-SAM | SAM | LE-SAM (Ours) |
|---|---|---|---|
| ResNet-50 | 76.93 | 76.70 | 77.41 |

| ImageNet | SGD | Eigen-SAM | ASAM | ESAM |
|---|---|---|---|---|
| ResNet-101 | 77.81 | 78.96 | 78.43 | 79.10 |

| | F-SAM | SAM | LE-SAM (Ours) |
|---|---|---|---|
| ResNet-101 | 78.72 | 78.54 | 79.31 |

*Table 3.* Test accuracy for fine-tuning ViT-B-16 pretrained on ImageNet-1K on CIFAR-10 and CIFAR-100.

| Method | CIFAR-10 | CIFAR-100 |
|---|---|---|
| SGD* | $98.08 \pm 0.11$ | $88.31 \pm 0.13$ |
| SAM | $98.28 \pm 0.13$ | $89.34 \pm 0.10$ |
| LE-SAM (Ours) | $98.72 \pm 0.08$ | $89.65 \pm 0.11$ |

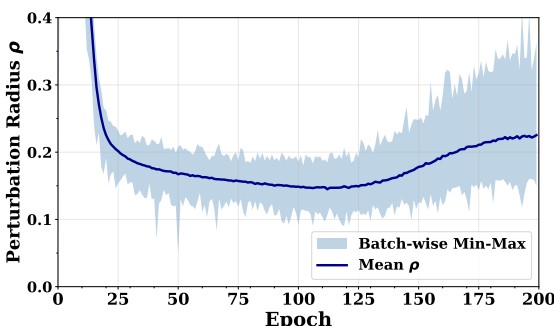

*Figure 5.* Perturbation radius across training process

empirically observe that our LE-SAM is able to seek flatter minima.

### 5.5. Mechanism Analysis

**Radius Dynamics under Loss-Budgeted Perturbations:** For our LE-SAM, the perturbation radius is not fixed; instead, it is back-solved from a fixed loss budget $\sigma$. The dynamic of the perturbation radius across the training process on CIFAR-100 with ResNet-18 is shown in Figure 5. When the gradient norm is small, the perturbation radius becomes large, as training progresses, the gradient scale grows,

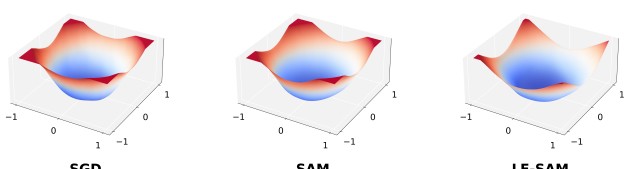

*Figure 4.* Visualization of loss landscape for SGD, SAM, and our LE-SAM

and the perturbation radius decreases, when optimization approaches a minimizer and gradients shrink again, $\rho$ rises at the late stage of training.

**Beyond Mainly Late-Stage Gains of SAM** Recent work on SAM points out that SAM works mainly in the late stage of training (Zhou et al., 2025). This phenomenon occurs since the practical surrogate of SAM contains a dominant gradient norm term, and only when gradients become small at the late stage of training does the curvature term becomes comparable. However, our LE-SAM fixes this at the mechanism level. By setting a fixed loss budget, the first-order variation becomes a constant, and its gradient with respect to the model parameters vanishes; the training signal is pushed toward the second-order term throughout training. Empirically, as shown in Figure 6, SAM shows only a small accuracy difference between SAM and SGD+SAM (switch at epoch 160), matching the observation that SAM mainly works in the late stage of training. In contrast, the performance gap between LE-SAM (without stabilization mechanisms) and SGD+LE-SAM (switch at epoch 160) is substantial, indicating that LE-SAM is not a mechanism that primarily takes effect in the late stage of training but rather one that remains effective throughout training.

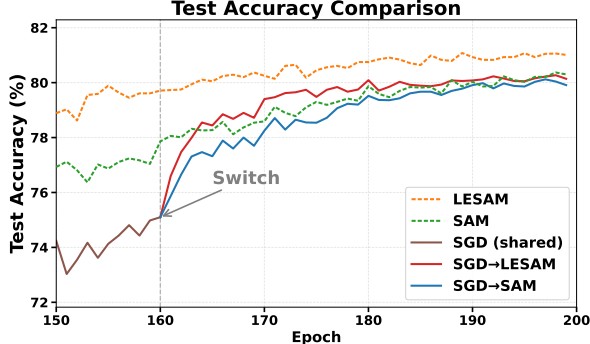

*Figure 6.* Test accuracy trajectories under whole run vs. late-stage switch training. We compare enabling SAM and LE-SAM throughout training against activating them after epoch 160 (SGD then SAM, SGD then LE-SAM). We apply all the methods to CIFAR-100 with ResNet-18.

| Model | SAM | LE-SAM | LE-SAM+ |
|---|---|---|---|
| ResNet-18 | 80.17 | 81.51 | **82.01** |
| WRN-28-10 | 83.42 | 84.91 | **85.06** |
| PyramidNet-110 | 84.46 | 85.91 | **86.19** |

*Table 4.* Test accuracy (%) comparison on CIFAR-100.

# 6. Additional Experiments and Analysis

## 6.1. LE-SAM+

From Eq. 13, since $\sigma$ is a scalar term and has zero gradient, the difference between $L_S(\mathbf{w} + \dot{\epsilon}_t)$ and $L_S(\mathbf{w})$ can be seen as a proxy to the second-order curvature term where:

$$\hat{H}(\mathbf{w}) = L_S(\mathbf{w}+\dot{\epsilon}_t) - L_S(\mathbf{w}) \approx \underbrace{\sigma}_{\text{scalar constant}} + \frac{1}{2}\dot{\epsilon}_t^\top H(\mathbf{w})\dot{\epsilon}_t. \quad (15)$$

Thus, we can extend the optimization objective to amplify the curvature awareness as follows:

$$L_S(\mathbf{w} + \dot{\epsilon}_t) + \alpha\hat{H}(\mathbf{w}). \quad (16)$$

Here $\alpha$ is a hyperparameter, we select $\alpha$ as 0.5 0.55 0.6 with corresponding $\sigma$ as 0.15 0.2 0.25 on three training backbones. We evaluate the LE-SAM+ performance on CIFAR-100, the experiment result is shown in Table 4, where the performance can be further improved.

## 6.2. Single Domain Generalization

In this section, we evaluate the generalization ability of our LE-SAM under the single domain generalization (SDG) experiment settings. In SDG settings, our aim is to train a model that can generalize well on different target domains with only one available source domain. we evaluate the performance on PACS (Li et al., 2017) (with pretrained ResNet-18) and VLCS (Fang et al., 2013) (with ResNet-18 without pretraining) with an improved baseline for domain

generalization ERM++ (Teterwak et al., 2025). The results are shown in Tab.1, LE-SAM can consistently surpass SAM by an average of 2.18 and 0.84 points on two datasets, demonstrating better generalization performance.

SDG Results on PACS

| Method | P | A | C | S | Avg |
|---|---|---|---|---|---|
| ERM++ | 43.82 | 65.60 | 74.32 | 35.38 | 54.78 |
| SAM | 44.87 | 65.09 | 74.11 | 38.69 | 55.69 |
| LE-SAM | 46.77 | 67.76 | 75.22 | 41.71 | 57.87 |

SDG Results on VLCS

| Method | V | L | C | S | Avg |
|---|---|---|---|---|---|
| ERM++ | 62.52 | 47.54 | 38.17 | 54.80 | 50.76 |
| SAM | 62.27 | 47.32 | 46.97 | 53.94 | 52.63 |
| LE-SAM | 62.39 | 48.30 | 47.65 | 55.54 | 53.47 |

## 6.3. Robustness Under Adversarial Attacks

We further evaluate the robustness of our method under adversarial attacks. Following (Zhang et al., 2024), we conduct experiments on CIFAR 10/100 with $L_2$ and $L_\infty$ PGD attacks (Madry et al., 2017) and compared with SAM. For SAM, we adopt suggested $\rho = 0.4$ following (Zhang et al., 2024), for LE-SAM, we choose 0.2, 0.4 for $\sigma$ on CIFAR-10 and CIFAR-100. Since the paper (zhang, 2024) and its workshop version did not mention the attack step size selection on the test set, we follow the hyperparameter selection in the eval.py in the paper's repository. The results are shown in Table 5, LE-SAM outperforms 2.44 0.79 2.47 and 0.90 points under all scenarios, showing stronger robustness under adversarial attacks, especially facing stronger attacks.

## 6.4. Further Experiments

To further evaluate the stability of our LE-SAM algorithm, we conduct a sensitivity analysis on our core hyperparameter $\sigma$; the experiment details are shown in Appendix Figure 7. Meanwhile, we also apply our LE-SAM under long-tailed training scenarios; the experiments are shown in Appendix Table 7. Notably, we achieve state-of-the-art performance on CIFAR-100-LT (Cui et al., 2019), showing the strong generalization performance of our algorithm. Moreover, we further analyze the computational complexity and evaluate the training time cost in Appendix C.

# 7. Related Work

**SAM and Flat Minima** The connection between the geometry of the loss landscape and the generalization capa-

| Method | CIFAR10 | | CIFAR100 | |
| | $\ell_\infty$-PGD $\epsilon = \frac{1}{255}$ | $\ell_2$-PGD $\epsilon = \frac{32}{255}$ | $\ell_\infty$-PGD $\epsilon = \frac{1}{255}$ | $\ell_2$-PGD $\epsilon = \frac{32}{255}$ |
| --- | --- | --- | --- | --- |
| SAM | 55.47 | 64.33 | 27.49 | 36.34 |
| LE-SAM | **57.91** | **65.12** | **29.86** | **37.24** |

*Table 5.* Robust accuracy evaluation on CIFAR-10/100 dataset.

bility of Deep Neural Networks is well-established. The influential flat minima hypothesis assumes that solutions residing in wide, low-curvature regions generalize better than those in sharp basins, a theory supported by both classical work (Hochreiter & Schmidhuber, 1994) and modern empirical studies linking sharp minima to the large-batch generalization gap (Keskar et al., 2017; Dinh et al., 2017).

Driven by this principle, Sharpness-Aware Minimization (SAM) (Foret et al., 2021) explicitly seeks flat minima by minimizing the worst-case loss within a local neighborhood (Zhang et al., 2024). While effective, SAM's reliance on a fixed Euclidean ball and its computational overhead have spurred a diverse body of research, ranging from universal frameworks (Tahmasebi et al., 2024) to domain generalization applications (Dong et al., 2024). To address the anisotropy of the parameter space, **geometric-aware variants** such as ASAM (Kwon et al., 2021) and Fisher-SAM (Kim et al., 2022) introduce adaptive normalization metrics (Zhou et al., 2022), while Eigen-SAM (Luo et al., 2025) explicitly aligns perturbations with the principal Hessian direction. Beyond geometry, other approaches focus on **optimization stability**, for instance, GSAM (Zhuang et al., 2022) and Friendly-SAM (Li et al., 2024) aim to filter out harmful sharpness signals or gradient noise to stabilize the trajectory, while other works propose weighted sharpness as an explicit regularizer (Yue et al., 2024). Simultaneously, to mitigate the computational cost, **efficiency-oriented methods** like ESAM (Du et al., 2022) and Momentum-SAM (Becker et al., 2025) employ strategies ranging from stochastic weight updates to historical gradient reuse.

Despite these extensive advancements, most existing variants continue to operate within the original mini-max framework, primarily refining the perturbation direction or efficiency under a *fixed-radius constraint*. In contrast, our work re-examines the foundational mechanism of SAM. We propose a novel *loss-equated* formulation that fundamentally rethinks how flatness is pursued, moving beyond the standard perturbation paradigm.

**Polyak Step-Size**  Polyak step-size (Polyak, 1987) is a classical adaptive step-size strategy. Instead of relying on a fixed learning rate schedule, its update magnitude is based on the current loss gap to the optimum, making the update magnitude primarily governed by the loss gap instead of the gradient scale. Recently, the Polyak step-size has

been revisited from a modern convex optimization perspective. Hazan & Kakade (2019) illustrate that the Polyak step-size can achieve near-optimal convergence rates without prior knowledge of smoothness or strong convexity constants. Follow-up theoretical work includes refined worst-case/complexity analysis under momentum or acceleration (Barré et al., 2020) and tight last-iterate convergence guarantees for nonsmooth convex subgradient methods using Polyak steps (Zamani & Glineur, 2024). Meanwhile, Polyak step-size has also been adapted to stochastic settings in modern stochastic optimization and deep learning. Loizou et al. (2021) proposed the Stochastic Polyak Step-size, replacing full gradient quantities by sample-wise evaluations. Subsequent work investigated the dynamic and stability of SGD under stochastic Polyak step-sizes (Orvieto et al., 2024) and explored combinations with line-search, preconditioning, and momentum to improve robustness and practicality (Jiang & Stich, 2023; Oikonomou & Loizou, 2024).

## 8. Conclusion

In this work, we revisit Sharpness-Aware Minimization (SAM) and identify a mechanism-level objective mismatch: while flat minima are fundamentally a second-order notion, practical SAM training is largely dominated by a first-order, gradient norm–driven surrogate induced by the fixed adversarial perturbation radius. To fix this, we propose Loss-Equated SAM (LE-SAM), which fixes a loss-space budget and back-solves the corresponding perturbation radius, effectively removing gradient-norm dominance and shifting the optimization signal toward curvature-dominated terms. LE-SAM consistently outperforms SAM and strong SAM variants, and our Hessian/landscape analysis further supports that LE-SAM biases optimization toward flatter solutions.

## Impact Statement

This paper presents work whose goal is to advance the field of Machine Learning. There are many potential societal consequences of our work, none of which we feel must be specifically highlighted here.

## Acknowledgments

The work was partially supported by the following: the Zhejiang Provincial Natural Science Foundation – Exploration Project under No. LMS26F020007, the Wenzhou Applied Fundamental Research Program (Basic Research) under No. GG20250198, the WKU 2026 International Frontier Interdisciplinary Research Institute Talent Program under No. WKUTP2026002, the WKU 2025 International Collaborative Research Program under No. ICRPSP2025001.

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

# A. Detailed Summary of Sharpness-Aware Minimization Variants

In this section, we will conduct an in-depth classification review of representative Sharpness-Aware Minimization (SAM) variants. We divide these developments into three main aspects: geometric refinement, optimization stability, and computational efficiency. This comprehensive comparison illuminates the evolution of sharpness-aware optimization and highlights the motivation behind LE-SAM, which fundamentally shifts the adversarial mechanism from a fixed-radius constraint in parameter space to a fixed-budget constraint in loss space.

Following our unified notation, let $\mathbf{w}$ denote the model parameters, $L_{\mathcal{S}}(\mathbf{w})$ the training loss on a mini-batch $\mathcal{S}$, and $\mathbf{g} = \nabla_{\mathbf{w}} L_{\mathcal{S}}(\mathbf{w})$ the vanilla gradient.

## A.1. Variants Focused on Neighborhood Geometry

**ASAM: Adaptive SAM (Kwon et al., 2021).** Standard SAM is sensitive to the rescaling of weights. For example, in a batch normalized layer, scaling weights does not change the output, but it does change the Euclidean distance. ASAM addresses this by introducing a parameter-wise normalization matrix $T_{\mathbf{w}} = \mathrm{diag}(|w_1|, \ldots, |w_d|)$, defining an adaptive ellipsoid neighborhood. The perturbation $\hat{\boldsymbol{\epsilon}}$ is:

$$\hat{\boldsymbol{\epsilon}}_{\mathrm{ASAM}} = \rho \frac{T_{\mathbf{w}}^2 \mathbf{g}}{\|T_{\mathbf{w}} \mathbf{g}\|_2}. \tag{17}$$

By ensuring scale invariance, ASAM provides a more consistent sharpness metric across different network architectures, though it still relies on a manually tuned fixed radius $\rho$.

**Eigen-SAM: Explicit Curvature Alignment (Luo et al., 2025).** Eigen-SAM argues that the standard SAM perturbation in the normalized gradient direction may be poorly aligned with the leading eigenvector of the Hessian, limiting its ability to regularize the top Hessian eigenvalue. It periodically estimates the leading eigenvector $\hat{\mathbf{v}}$ of the Hessian matrix $\mathbf{H} = \nabla^2 L_{\mathcal{S}}(\mathbf{w})$ and decomposes it into components parallel and perpendicular to the gradient direction:

$$\hat{\mathbf{v}} = \hat{\mathbf{v}}_{\|} + \hat{\mathbf{v}}_{\perp}. \tag{18}$$

The perturbation direction is then modified as:

$$\boldsymbol{\epsilon}_{\mathrm{Eigen}} = \frac{\mathbf{g}}{\|\mathbf{g}\|_2} + \alpha \, \mathrm{sign}(\langle \mathbf{g}, \hat{\mathbf{v}} \rangle) \hat{\mathbf{v}}_{\perp}, \quad \tilde{\mathbf{w}} = \mathbf{w} + \rho \boldsymbol{\epsilon}_{\mathrm{Eigen}}. \tag{19}$$

Here $\alpha$ controls the strength of explicit alignment. Thus, Eigen-SAM preserves the SAM gradient-based perturbation while adding the orthogonal component of the estimated leading eigenvector to improve curvature alignment.

**Fisher-SAM (Kim et al., 2022).** The Fisher-SAM algorithm utilizes the Fisher Information Matrix (FIM) as a metric tensor to define the neighborhood. Instead of Euclidean distance, it considers the information-geometric distance (KL-divergence) between the original and perturbed models. This approach ensures that the perturbation respects the underlying geometry of the probability manifold for smoothing the loss landscape.

## A.2. Variants Focused on Optimization Stability

These methods are designed to alleviate the instability or the gradient noise introduced by the adversarial ascent steps.

**GSAM: Surrogate Gap Minimization (Zhuang et al., 2022).** GSAM identifies that simply minimizing the perturbed loss $L_{\mathcal{S}}(\mathbf{w} + \hat{\boldsymbol{\epsilon}})$ does not necessarily reduce sharpness if the original loss $L_{\mathcal{S}}(\mathbf{w})$ decreases faster. It defines a surrogate gap $h(\mathbf{w}) = L_{\mathcal{S}}(\mathbf{w} + \hat{\boldsymbol{\epsilon}}) - L_{\mathcal{S}}(\mathbf{w})$ and decomposes the update into a component that minimizes the perturbed loss and a component that minimizes this gap. Crucially, it computes the original gradient $\mathbf{g} = \nabla L_{\mathcal{S}}(\mathbf{w})$ and the perturbed gradient $\hat{\mathbf{g}} = \nabla L_{\mathcal{S}}(\mathbf{w} + \hat{\boldsymbol{\epsilon}})$. It then projects the **original gradient g** onto the direction **orthogonal to the perturbed gradient $\hat{\mathbf{g}}$** to obtain a sharpness-reducing component $\mathbf{g}_{\perp}$:

$$\mathbf{g}_{\perp} = \mathbf{g} - \frac{\mathbf{g}^{\top} \hat{\mathbf{g}}}{\|\hat{\mathbf{g}}\|^2} \hat{\mathbf{g}},$$
$$\mathbf{g}_{\mathrm{GSAM}} = \hat{\mathbf{g}} - \alpha \mathbf{g}_{\perp}, \tag{20}$$

where $\alpha$ is a hyperparameter controlling the strength of surrogate gap minimization.

**SSAM: Stabilizing via Renormalization (Tan et al., 2024).** SSAM addresses the training instability in SAM caused by the drastic variance of gradient magnitudes between the original and perturbed points. It stabilizes the optimization by a simple renormalization strategy: before the descent step, the gradient at the perturbed point $\hat{\mathbf{g}}$ is rescaled to match the norm of the original gradient $\mathbf{g}$. The update direction is therefore:

$$\mathbf{g}_{\text{SSAM}} = \frac{\|\mathbf{g}\|_2}{\|\hat{\mathbf{g}}\|_2}\hat{\mathbf{g}}. \tag{21}$$

This ensures $\|\mathbf{g}_{\text{SSAM}}\|_2 = \|\mathbf{g}\|_2$, effectively preventing the erratic updates that can hinder training and allowing for a more stable and consistent optimization path.

**Friendly-SAM (F-SAM) (Li et al., 2024).** F-SAM investigates the components within SAM's adversarial perturbation and finds that the stochastic gradient noise, not the full gradient, is the key to generalization improvement. To harness this, F-SAM explicitly removes the estimated full gradient component and uses the remaining stochastic gradient noise to construct a more friendly perturbation. In a simplified form, it approximates the full gradient via an Exponential Moving Average (EMA) of historical gradients and computes the perturbation as:

$$\hat{\boldsymbol{\epsilon}}_{\text{F-SAM}} = \rho\frac{\mathbf{g} - \mathbf{g}_{\text{EMA}}}{\|\mathbf{g} - \mathbf{g}_{\text{EMA}}\|_2}, \tag{22}$$

where $\mathbf{g}$ is the current mini-batch gradient and $\mathbf{g}_{\text{EMA}}$ is the EMA estimate serving as a surrogate for the full gradient. This design encourages the ascent step to focus more on the batch-specific noise, enhancing the consistency and effectiveness of sharpness minimization.

### A.3. Efficient and Single-Step Variants

**ESAM: Efficient SAM (Du et al., 2022).** ESAM incorporates two key techniques: Stochastic Weight Perturbation (SWP) and Sharpness-sensitive Data Selection (SDS). SWP approximates the weight perturbation by perturbing a stochastically selected subset of parameters, while SDS selects a subset of sharpness-sensitive samples for the perturbed-loss update, thereby reducing the computational overhead of SAM.

**Momentum-SAM (MSAM) (Becker et al., 2025).** Directly inspired by Nesterov Accelerated Gradient (NAG), MSAM eliminates the extra forward-backward pass by repurposing the existing momentum vector. It uses the momentum buffer to construct a perturbation in the negative momentum direction, achieving sharpness-aware optimization without any additional gradient computation, thus matching the per-iteration cost of the base optimizer.

**S2-SAM: Single-step SAM (Ji et al., 2024).** This variant is designed to compute a SAM-style update with only one forward-backward pass per iteration. Instead of computing an additional gradient to form the current perturbation direction, S2-SAM reuses the previous-step gradient to construct the perturbed point, and then evaluates the gradient once at that perturbed point. In this way, it reduces the extra computational cost of vanilla SAM while still retaining a sharpness-aware update.

## B. Mathematical Derivations for LE-SAM

### B.1. Polyak-Informed Derivation of the Loss-Equated Radius

The core mechanism of LE-SAM is to determine the perturbation radius $\rho_t$ such that the first-order loss variation approximately matches a fixed budget $\sigma$. We begin with the first-order Taylor expansion:

$$L_{\mathcal{S}}(\mathbf{w} + \boldsymbol{\epsilon}) - L_{\mathcal{S}}(\mathbf{w}) \approx \mathbf{g}^{\top}\boldsymbol{\epsilon}. \tag{23}$$

Assuming the perturbation is aligned with the gradient direction $\boldsymbol{\epsilon} = \rho\frac{\mathbf{g}}{\|\mathbf{g}\|_2}$, we set the linear variation to $\sigma$:

$$\mathbf{g}^{\top}\left(\rho\frac{\mathbf{g}}{\|\mathbf{g}\|_2}\right) = \sigma \implies \rho\|\mathbf{g}\|_2 = \sigma \implies \rho_t = \frac{\sigma}{\|\mathbf{g}\|_2}. \tag{24}$$

**Remark (Numerical Stability):** In our practical implementation (Algorithm 1), we introduce a small stability constant $\varrho$ to the denominator, i.e., $\rho_t = \sigma/(\|\mathbf{g}\|_2 + \varrho)$, to prevent numerical instability when the gradient norm approaches zero. In

the idealized case without the stabilizer, substituting this adaptive radius $\rho_t$ back into the second-order expansion of the perturbed loss gives:

$$L_{\mathcal{S}}(\mathbf{w} + \hat{\epsilon}) \approx L_{\mathcal{S}}(\mathbf{w}) + \underbrace{\sigma}_{\text{constant}} + \frac{1}{2}\hat{\epsilon}^\top \mathbf{H}\hat{\epsilon}. \tag{25}$$

Since $\nabla_{\mathbf{w}}\sigma = 0$, the first-order variation is effectively scalarized into a constant budget. This eliminates the gradient-norm-dominated component from the optimization signal, forcing the model to minimize the second-order curvature term $\frac{1}{2}\hat{\epsilon}^\top \mathbf{H}\hat{\epsilon}$ directly throughout training.

### B.2. Optimality of the Loss-Equated Perturbation

We seek the minimum-norm perturbation $\epsilon$ that satisfies a fixed loss increase $\sigma$ in the loss space:

$$\min_{\epsilon} \frac{1}{2}\|\epsilon\|_2^2 \quad \text{s.t.} \quad \mathbf{g}^\top \epsilon = \sigma.$$

The Lagrangian is $\mathcal{L}(\epsilon, \lambda) = \frac{1}{2}\|\epsilon\|_2^2 - \lambda(\mathbf{g}^\top \epsilon - \sigma)$. The optimality conditions are:

$$\nabla_\epsilon \mathcal{L} = \epsilon - \lambda \mathbf{g} = 0 \implies \epsilon = \lambda \mathbf{g}.$$

Plugging into the constraint: $\mathbf{g}^\top(\lambda \mathbf{g}) = \sigma \implies \lambda = \sigma/\|\mathbf{g}\|_2^2$. Thus, the optimal perturbation is $\epsilon^* = \frac{\sigma}{\|\mathbf{g}\|_2^2}\mathbf{g}$. This result aligns with our radius formulation in Algorithm 1, confirming that scaling the **unit gradient** $(\mathbf{g}/\|\mathbf{g}\|_2)$ by $\rho_t \approx \sigma/\|\mathbf{g}\|_2$ (with $\rho_t = \sigma/(\|\mathbf{g}\|_2 + \varrho)$ in practice for numerical stability) represents the minimum-norm perturbation required to satisfy the loss budget $\sigma$.

## C. Computational Complexity Analysis

LE-SAM maintains the same asymptotic computational complexity as standard SAM, requiring exactly two forward-backward passes per iteration. The additional overhead for LE-SAM lies in the calculation of the dynamic radius $\rho_t = \sigma/(\|\mathbf{g}\|_2 + \varrho)$. This involves a vector norm computation $(O(d))$ and a few scalar operations $(O(1))$, where $d$ denotes the total number of model parameters. We compare the training time of LE-SAM with standard SAM using the same benchmark script and the original training loops of both methods shown in Table 6. All experiments are conducted on a single NVIDIA RTX 5090 (32 GB) GPU.

*Table 6.* Training time per epoch of SAM and LE-SAM measured by directly reusing their original two-pass CIFAR-100 training loops. For each architecture, we run 5 epochs in total, and report the average time.

| Architecture | Optimizer | Time / Epoch (s) |
|---|---|---|
| ResNet-18 | SAM | 9.06 |
| ResNet-18 | LE-SAM | 9.32 |
| WideResNet-28-10 | SAM | 55.83 |
| WideResNet-28-10 | LE-SAM | 56.26 |
| PyramidNet | SAM | 88.82 |
| PyramidNet | LE-SAM | 89.72 |

## D. Sensitivity Analysis

We analyze the sensitivity of LE-SAM to its core hyper-parameter: the loss budget $\sigma$. We evaluate the sensitivity of the loss budget $\sigma$ by varying it from 0.3 to 0.5 on CIFAR-100 with ResNet-18. As the result shown in Figure 7, the performance remains relatively stable across a wide range of $\sigma$ values. Specifically, smaller $\sigma$ weakens the adversarial perturbation effect, making the method closer to standard SGD behavior, while an overly large $\sigma$ may introduce excessive perturbations and harm optimization stability.

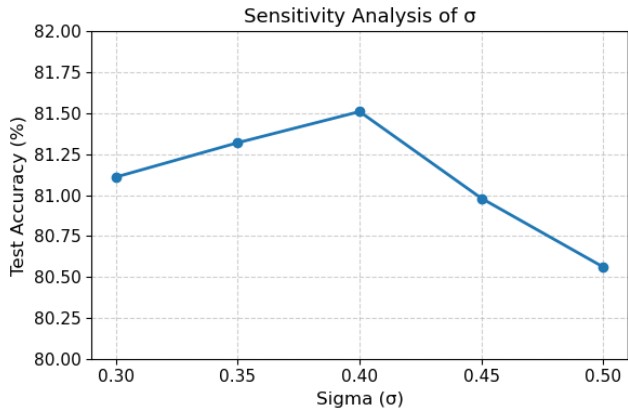

*Figure 7.* Sensitivity analysis of loss budget $\sigma$

*Table 7.* Results on CIFAR-100-LT with different imbalance factors.

| Method | CIFAR-100-LT | |
|---|---|---|
| | 100 | 50 |
| CE | 42.1 | 46.1 |
| BBN (CVPR20) | 42.6 | 47.1 |
| KCL (ICLR21) | 42.8 | 46.3 |
| TSC (CVPR22) | 43.8 | 47.4 |
| HCL (CVPR21) | 46.7 | 51.9 |
| ETF-DR (NeurIPS22) | 45.3 | 50.4 |
| SEL (ICCV25) | 44.5 | 49.6 |
| Focal-SAM (ICML25) | 44.0 | 48.1 |
| RBL (ICML23) | 53.1 | 57.2 |
| GLMC | 55.9 | 61.1 |
| GLMC + SAM | $57.25_{\pm 0.34}$ | $63.01_{\pm 0.41}$ |
| GLMC + LE-SAM(Ours) | $58.65_{\pm 0.46}$ | $63.67_{\pm 0.47}$ |

## E. Imbalance Learning

Real-world data always imbalanced and follow the long-tailed distribution. Under the long-tailed distribution, since the sample space is dominated by the head classes, the model tends to overfit or favor head classes while under-performing on tail classes. Following the standard imbalance-learning protocol, we conduct our experiments on the CIFAR-100-LT dataset (Cui et al., 2019) with two imbalance factors {100, 50} (The imbalance ratio is defined as the ratio between the number of samples in the most frequent (head) class and that in the least frequent (tail) class). We choose various long-tail methods as experiment baselines including traditional approaches and recently proposed method with the highest performance (Zhou et al., 2020; Kang et al., 2020; Li et al., 2022; Wang et al., 2021; Yang et al., 2022; Jian et al., 2025; Li et al., 2025; Peifeng et al., 2023; Du et al., 2023).

As the experiment results shown in Table 7, when combining our LE-SAM with the current SOTA method GLMC, the performance can be further improved. Specifically, our approach outperforms GLMC 2.75 and 2.57 points on two different imbalance ratios and achieves the SOTA performance while consistently outperforming GLMC with standard SAM, demonstrating the strong generalization performance of our loss-equated adversarial perturbation mechanism.

## F. Additional Experimental Details

**Hardware.** All experiments were conducted on a cluster equiped with NVIDIA GeForce RTX 4090 and 5090 GPUs.

**Datasets** We evaluate LE-SAM on CIFAR-10, CIFAR-100 (Krizhevsky et al., 2009), and ImageNet-1K (Deng et al., 2009). For CIFAR, we use ResNet-18 (He et al., 2016), WideResNet-28-10 (Zagoruyko & Komodakis, 2016), and PyramidNet-110

(Han et al., 2017), training for 200 epochs. We use an SGD optimizer with a momentum of 0.9 and a weight decay of 5e-4. Our implementation is based on the open-source repository of ESAM (Du et al., 2022).

**Hyper-parameters.** The detailed hyperparameter selections are shown in Table 8 and Table 9.

*Table 8.* Training hyperparameters for CIFAR-10 and CIFAR-100 using LE-SAM.

| Model | CIFAR-10 (LE-SAM) | CIFAR-100 (LE-SAM) |
| --- | --- | --- |
| **ResNet-18** | | |
| Epoch | 200 | 200 |
| Batch size | 128 | 128 |
| Data augmentation | Basic | Basic |
| Peak learning rate | 0.05 | 0.05 |
| Learning rate decay | Cosine | Cosine |
| Weight decay | $1 \times 10^{-3}$ | $1 \times 10^{-3}$ |
| $\rho_{max}$ | 0.3 | 0.3 |
| $\sigma$ | 0.35 | 0.35 |
| Budget annealing (epoch) | 160 | 160 |
| **Wide-28-10** | | |
| Epoch | 200 | 200 |
| Batch size | 128 | 128 |
| Data augmentation | Basic | Basic |
| Peak learning rate | 0.05 | 0.05 |
| Learning rate decay | Cosine | Cosine |
| Weight decay | $1 \times 10^{-3}$ | $1 \times 10^{-3}$ |
| $\rho_{max}$ | / | / |
| $\sigma$ | 0.3 | 0.3 |
| Budget annealing (epoch) | / | / |
| **PyramidNet-110** | | |
| Epoch | 300 | 300 |
| Batch size | 256 | 256 |
| Data augmentation | Basic | Basic |
| Peak learning rate | 0.10 | 0.10 |
| Learning rate decay | Cosine | Cosine |
| Weight decay | $5 \times 10^{-4}$ | $5 \times 10^{-4}$ |
| $\rho_{max}$ | / | / |
| $\sigma$ | 0.3 | 0.3 |
| Budget annealing (epoch) | / | / |

*Table 9.* Hyperparameters for training on ImageNet using LE-SAM.

| Hyperparameter | ResNet-50 (LE-SAM) | ResNet-101 (LE-SAM) |
|---|---|---|
| Epoch | 90 | 90 |
| Batch size | 512 | 512 |
| Peak learning rate | 0.2 | 0.2 |
| Learning rate decay | Cosine | Cosine |
| Weight decay | $1 \times 10^{-4}$ | $1 \times 10^{-4}$ |
| Input resolution | $224 \times 224$ | $224 \times 224$ |
| $\rho_{max}$ | 0.2 | 0.2 |
| $\sigma$ | 0.3 | 0.3 |
| Budget annealing (epoch) | 60 | 60 |

