# OpenReview forum: "Fix the Loss, Not the Radius: Rethinking the Adversarial Perturbation of Sharpness-Aware Minimization"
_ICML.cc/2026/Conference — ICML 2026 regular_

### Official Review · Reviewer_cxtx · 2026-03-10

**Soundness:** 3
**Presentation:** 2
**Significance:** 3
**Originality:** 4
**Overall Recommendation:** 5
**Confidence:** 5

**Summary:**

In this paper, the authors have proposed Loss-Equated SAM (LE-SAM), which highlights the blind spot of SAM, where SAM's update is dominated by the first-order gradient rather than the curvature; thus, blurring its goal of reaching a low-curvature loss surface. From this perspective, LE-SAM zeroes the dominance of gradient norm by simply fixing it as a constant, which naturally steers SAM to focus on the second-order derivative. In the popular image classification benchmarks, including CIFAR and ImageNet, LE-SAM slightly outperforms SAM and its variants.

**Compliance With Llm Reviewing Policy:**

Affirmed.

**Final Justification:**

All concerns are resolved. The main changes are about additional flatness measurements and OOD testing. All results support the strength of the proposed method, LESAM. I believe that the authors' technical proposal is not merely an incremental technical improvement, because LESAM can resolve the task- and model-wise exhaustive effort required to determine the perturbation radius when applying SAM, thereby broadly innovating in flat-minima search (finding loss budget seems to be easier).

Based on this assessment, I hope to raise my score to 'Accept' from 'Weak Accept'.

**Key Questions For Authors:**

**Question 1: Asking for the key difference of LE-SAM beyond SAM**
- I tried to figure out how LE-SAM leads to a different optimization trajectory than SAM.
- When assumed to start an update at the same weight $w$, SAM and LE-SAM compute the same perturbation direction $g_c$.
- To my understanding, the key difference comes from the different derivatives of $L_N(w)$. Specifically, LE-SAM zeroes the first-order term dominated by the norm of the gradient (i.e., scalar constant in Eq. (13)), thereby focusing on the curvature to get the update direction. Therefore, this gap leads to a different (hopefully flatter) minimum with SAM.
- Would you clarify whether my understanding is correct or not?

**Question 2: Asking for the additional flatness metrics**
- Would you provide additional quantitative flatness metrics, including LPF (Bisla, 2022) and $\mu_{PAC-Bayes}$ (Jiang, 2020)?

> D. Bisla et al., Low-Pass Filtering SGD for Recovering Flat Optima in the Deep Learning Optimization Landscape, AISTATS 2022.

> Y. Jiang, Y et al., Fantastic generalization measures and where to find them. ICLR 2020.

**Question 3: Asking for out-of-distribution testing, e.g., adversarial robustness and domain generalization**
- Would you provide additional empirical results on the out-of-distribution testing?
- For adversarial robustness, it will be proper to follow the adversarial robustness settings in (Zhang, 2024) for CIFAR-10/100 cases. For ImageNet, it seems tough to do it in the limited time during the rebuttal. Please do CIFAR-10 or CIFAR-100 tests with $L_2$ and $L_{\infty}&-PGD attacks. Testing both datasets will be highly appreciated.
- For domain generalization, please refer to DomainBed (Gulrajani, 2021) benchmarks. I guess that not all benchmarks are needed, but LE-SAM on PACS and VLCS would be fine.

> Y. Zhang et al., On the Duality Between Sharpness-Aware Minimization and Adversarial Training, ICML 2024.

> I. Gulrajani and D. Lopez-Paz, In Search of Lost Domain Generalization, ICLR 2021.

**Limitations:**

I think that this paper shows a few limitations, but they are not stated in the submitted manuscript. Specifically,
- A limited evaluation of out-of-distribution testing, including adversarial robustness and domain generalization. There exist many other benchmarks for evaluating the generalization capability of SAM and SAM-variant. I have suggested further providing two basic out-of-distribution tests.
- More quantitative flatness metrics are needed to fully guarantee the better flatness obtained by LE-SAM

**Strengths And Weaknesses:**

**Strength 1 (*Soundness and Originality*): A principled remedy to focus on the second-order term in flat minima searching**
- The main strength of this paper comes from the simple yet rigorous grounds of the suggested idea. It is not new to acknowledge that SAM's technical steps rely heavily on the first-order term; thus, SAM's efficacy generally appears at the intermediate-later stage of training. However, SAM's powerful/general performance across tasks/architecture mostly blinds us to this issue.
- The authors formally point out this issue, and derive a simple yet principled recipe to totally focus on the second-order derivative term in flatter minima searching (referring to Sections 4.1 and 4.2) *I believe that this direction sounds well, and applies to the variety of prior SAM-based methods.*
- To my understanding, Eq. (13) formally proves that the updates of LE-SAM lead to the optima, which prioritizes the second-order term by dropping the to-be-constant term from the gradient-norm (i.e., 'scalar constant' in Eq. (13)).

**Strength 2 (*Significance*): A broad applicability to flat minima searching applications**
- SAM-based flat minima searching is widely accepted as a model-agnostic solution to enhance model robustness, even across diverse tasks. Its applicability includes adversarial training, reinforcement learning, generative models, etc.
- By innovating SAM-variant flat minima approaches, I expect that LE-SAM shows an impact on various ML applications aiming at improving model robustness.

**Weakness 1 (*Soundness*): Additional flatness metrices should be shown.**
- As quantitative measurements of flatness, Hessian-based values, i.e., $\lambda_{max}$ and *Trace*, are measured (in Figure 3). It demonstrates that LE-SAM effectively normalizes large Hessian evaluations.
- They are popular and traditional metrics for quantifying flatness, but it has been frequently argued that these are sometimes shaky in representing the flatness and generalization; thus, other complementary metrics have been suggested. I recommend adding more metrics.
- For example, LPF (Bisla et al., 2022) measures the smoothness of the loss surface by employing a Gaussian kernel. In addition, PAC-Bayes measurement, i.e., $\mu_{PAC-Bayes}$ (Jiang et al., 2020), provides PAC-based flatness evaluations.

> D. Bisla et al., Low-Pass Filtering SGD for Recovering Flat Optima in the Deep Learning Optimization Landscape, AISTATS 2022.

> Y. Jiang, Y et al., Fantastic generalization measures and where to find them. ICLR 2020.

**Weakness 2 (Soundness): Missing generalization testing**
- The key benefit of finding flatter minima is to bolster the generalization capability of deep models.
- I acknowledge that the test accuracies of in-distribution (e.g., ImageNet, CIFAR) are the simple measurements of generalization.
- However, SAM's potential and benefits have been extensively confirmed in out-of-distribution testing, including adversarial robustness (Zhang, 2024), domain generalization (Cha et al., 2021), and even reinforcement learning (Lee et al., 2025).
- It does not need to move far, but I think that the demonstration of LE-SAM on adversarial robustness and domain generalization are required. If possible to add new results, I am quite sure that LE-SAM consistently maintains generalization and even further improves SAM across various out-of-distribution cases (please see **Question** for asking additional tests during rebuttal period).

> Y. Zhang et al., On the duality between sharpness-aware minimization and adversarial training, ICML 2024.

> J. Cha et al., SWAD: Domain Generalization by Seeking Flat Minima, NeurIPS 2021.

> H. K Lee et al., Flat Reward in Policy Parameter Space Implies Robust Reinforcement Learning, ICLR 2025.

**Weakness 3 (*Significance*): Marginal gains over SAM and SAM-variants**
- In the evaluations on CIFAR and ImageNet, the gains of LE-SAM look marginal beyond SAM and SAM-variants (mostly less than 1%).
- I am quite worried that LE-SAM's mathematical motivation is strong, but its practical impact in bolstering generalization may be minimal.
- I think that there exist many out-of-distribution testing, where the generalization effects become amplified, thus suggesting the follow-up out-of-distribution testing during the rebuttal period (please see **Question**).

**Minor weakness 4 (*Presentation*): Some minor corrections**
- An error in the second reference titled *Some large-scale matrix computation problems. Journal of Computational and Applied Mathematics*. The second author's name should be "Fahey, M", rather than "Fahey, G". It seems a clear *human error*. Because, in the BibTeX of the official publication, it says "Fahey, Gark", but it mismatches the printed name in the manuscript, which is "Fahey, Mark". From the Scopus information, the author, "Fahey, Mark," looks correct. I suggest that the authors revise it.
- The beginning paragraph in Section 3.1 looks redundant. At that point, it would be better to directly dive into the formal description of the SAM objective.
- Please use "Equation (2)" rather than "Equation 2".
- When referring to equations, please use the consistent notation between Eq. and Equation.
- At line 270, please add a whitespace after $\sigma$
- For illustrative presentations, most of the figures should be higher resolution without slight blurring.

---

> ### Author Rebuttal · Authors · 2026-03-31
>
> Our supplementary experimental results are available in the anonymous repository: https://anonymous.4open.science/r/rebuttalsam-F7B0. The figures and tables referred to in the rebuttal correspond to those in the repository.
>
>
> Q1: The reviewer's intuition is largely correct. More precisely, the key difference is in the mechanism level that SAM uses a fixed radius $\rho$, while LE-SAM fixes a loss budget
> $\sigma$ and back-solves a corresponding ideal radius. In addition, LE-SAM turns the first-order loss variation into a constant, so this term no longer dominates the outer optimization. As a result, the effective learning signal becomes more curvature-dominated.
>
> W1 and Q2 (Additional flatness metric): We evaluate $\mu_{pac-bayes}$ and LPF as additional metric. For $\mu_{pac-bayes}$ we follow eq.48 in (Jiang, 2020), we set the increase in training loss to a target value of 0.1, the final value for SAM is $5.8\times10^{8}$, LE-SAM is $3.2\times10^{8}$. For LPF, we track the value for both SAM and LE-SAM across 200 epochs shown in Fig.3 (Following (Bisla, 2022), we approximated LPF
> using the MC method.), LE-SAM has consistently lower LPF value. Overall, both two additional metric $\mu_{pac-bayes}$ and LPF, the hessian results and the visualization in main paper all indicates LE-SAM is able to seek a much flatter minima.
>
> W2 and Q3 (OOD): We appreciate the reviewer’s suggestion, and we agree that including more OOD evaluations is meaningful. Thus, we evaluate single domain generalization (SDG) performance (using one domain as the source domain and generalize on the rest domains) and  robustness under adversarial attack of LE-SAM. For SDG, we evaluate the performance on PACS (pretrained ResNet-18) and VLCS (ResNet-18 without pretraining) with a improved baseline for domain generalization ERM++ (Teterwak, 2025). The results are shown in Tab.1, LESAM can consistently surpass SAM an average of 2.18 and 0.84 points on two datasets, demonstrating better generalization performance. For robustness under advesarial attacks, we follow (zhang, 2024) for CIFAR 10/100 with $L_2$ and $L_{\infty}$ PGD attacks and compared with SAM. For SAM, we adopt suggested $\rho=0.4$ following (Zhang, 2024), for LE-SAM, we choose 0.2, 0.4 for $\sigma$ on CIFAR-10 and CIFAR-100. Since the paper (zhang, 2024) and its workshop version did not mentioned the attack step size selection on the test set, we follow the hyperparameter selection in the eval.py in the paper's repository. The results are shown in Tab.2, LE-SAM can outperform 2.44, 0.79, 2.47 and 0.90 points under all scenarios, showing stronger robustness under adversarial attacks, especially facing stronger attacks.
>
> W3: While the absolute gains may appear modest, we would like to emphasize that LE-SAM is a mechanism-level reformulation of SAM, rather than introducing additional regularization terms or extra modules. Unlike many SAM variants, LE-SAM only redesigns the perturbation mechanism while keeping the overall framework unchanged. Therefore, achieving consistent improvements over SAM and its variants without adding extra complexity or components demonstrates that the proposed mechanism is both principled and practically effective. Meanwhile, motivated by the reviewers' concern on annealing and clipping, we turn off the clipping and annealing mechanism and
> conduct the ablation. The results are shown in Tab.4. We found when we use relatively smaller $\sigma$ to ensure the convergence of the training, on WRN-28-10 ($\sigma=0.3$) and PyramidNet-110 ($\sigma=0.3$) the model performance can be further improved (from 84.43 to 84.91, from 85.72 to 85.91), we provide the loss/acc curve across training in Fig.5; Moreover, inspired by reviewer 1CdC Question2, we extend LESAM to LESAM+ which can further improve the generalization performance.
>
> LESAM+: From eq.13 in main paper, since $\sigma$ is a scaler term and has zero gradient, the difference between $L_S(\mathbf{w} + \dot{\epsilon_t} )$ and $L_S(\mathbf{w})$ can be seen as a proxy to the second-order curvature term:
> \begin{equation}
>  \hat{H}(\mathbf{w})= L_S(\mathbf{w} + \dot{\epsilon_t} )- L_S(\mathbf{w})
> \approx
> \underbrace{\sigma}_{\text{scalar constant}}+
> \frac{1}{2}\ \dot{\epsilon}_t^\top H(\mathbf{w})\ \dot{\epsilon}_t .
> \end{equation}
> Thus, we can extend the optimization objective to $L_S(\mathbf{w} + \dot{\epsilon_t} ) + \alpha \hat{H}(\mathbf{w})$ to amplify the curvature awareness where $\alpha$ is a hyperparameter, we select $\alpha$ as 0.5, 0.55, 0.6 with corresponding $\sigma$ as 0.15, 0.2, 0.25 on three training backbones. We evaluate the LESAM+ performance on CIFAR-100, the experiment result is shown in Tab.3, where the performance can be further improved (+0.68, +0.63, +0.47 on three training backbones). We also provide the loss/acc curve across training in Fig.2.
>
> Minor weaknesses: We will fix all the problems in the revised version.

---

> > ### Author Rebuttal · Reviewer_cxtx · 2026-04-01
> >
> > Thanks for the effort to relieve my concerns. **Two major concerns, i.e., flatness measurements and OOD testing, seem to be well resolved.** Consistent tendencies from the additional two flatness metrics support the flatter minima via LESAM. Also, OOD testing, particularly in domain generalization tasks, strongly confirms the enhanced generalization performance. For the marginal gain of LESAM, I appreciate the further technical optimization of LESAM+. In my opinion, regardless of the amount of gains, I believe that LESAM can relieve the exhaustive technical labor to optimize the perturbation radius $\rho$ for SAM (as acknowledged in Strength 2).
> >
> > Based on the assessment, I will raise my score to 'Accept', while keeping the highest confidence.

---

> > > ### Author Response · Authors · 2026-04-01
> > >
> > > Thank you very much for your recognition of our work and valuable suggestions！
> > >
> > >
> > > ——————Overall response: Thanks the Reviewers for their time and constructive feedback.————————
> > >
> > >
> > > We sincerely thank the reviewer cxtx for the constructive and detailed feedback.
> > >
> > >  The main concerns raised focused on (1) the lack of additional flatness metrics to more comprehensively validate the whether can seek a flatter minima. (2) out-of-distribution evaluations such as robustness under adversarial attacks and domain generalization (3) the concern that the empirical gains over SAM and its variants appear marginal.
> > >
> > > For(1) we addressed these concerns by adding additional flatness metrics, including LPF and PAC-Bayes-based measures, which consistently support that LE-SAM achieves flatter minima.
> > >
> > > For(2)  We further expanded the experimental evaluation to include single domain generalization (PACS, VLCS) with both pretraining and without pretraining training backbones and adversarial robustness under PGD attacks on CIFAR, where LE-SAM demonstrates consistent improvements over SAM.
> > >
> > >  For(3) we emphasized that LE-SAM is a mechanism-level reformulation without introducing extra components, and showed through ablations and the LESAM+ extension that the method can yield further improvements.
> > >
> > >  Overall, we thank the reviewer the reviewer cxtx for their valuable suggestions and positive score with the highest confidence . **Sincerely wish you all the best in your submissions and future research**.

---

### Official Review · Reviewer_R7nK · 2026-03-12

**Soundness:** 3
**Presentation:** 2
**Significance:** 3
**Originality:** 3
**Overall Recommendation:** 5
**Confidence:** 3

**Summary:**

Based on the observation that SAM's first-order surrogate is heavily influenced by the gradient norm, this paper introduces a new variant of SAM LE-SAM that, instead of finding the worst-case perturbation within a fixed radius ball, backsolves a fixed-loss budget to find the appropriate radius. The attempt removes the noisy gradient signal and thus focuses only on the curvature information and hence obtains flatter solutions. Experiments demonstrate that LE-SAM achieves better results compared to various SAM variants and achieves better sharpness at the obtained solution.

**Compliance With Llm Reviewing Policy:**

Affirmed.

**Final Justification:**

The rebuttal has addressed most of my concerns

**Key Questions For Authors:**

1. The annealing schedule is motivated by instability near convergence, but it also turns off the sharpness pressure entirely near minima rather than maintaining it in a stable form. Since SAM is well-behaved precisely when gradients are small, have you considered replacing the annealing schedule with a fallback to standard SAM when the dynamic radius exceeds the clipping cutoff? This would maintain sharpness pressure throughout convergence and also avoid introducing new hypers.

**Limitations:**

Not thoroughly discussed.

**Strengths And Weaknesses:**

Strengths:
1. The proposed modification of SAM is lightweight and does not require second-order computation, which makes the method very adaptable and reproducible.
2. Fig 3 empirically supports the effectiveness of LE-SAM in finding flatter minima, quantitatively supporting the paper’s flatness narrative.
3. The connection of SAM and the classical Polyak step-size is interesting.

Weaknesses:
1. The illustrative Fig. 2 could use some refining.
2. The analysis of SAM in section 3.3 shows that at local minima, the leading term that governs how fast the loss increases is the curvature, trivially true, and then concludes that flat minima are all about the Hessian spectrum. But here are two notions of sharpness that are considered: one is the sharpness of the minima found by the optimizer, which is precisely what is described here; the other is the information used during the optimization process, guiding towards the flat minima, in which the gradient still plays a non-trivial role. Indeed, second-order information became dominant at later stages of SAM as we move towards the center of the basin. However, the gradient norm dominance is a training dynamics observation and should not be claimed as a fundamental flaw or mismatch in SAM's definition of flatness.
3. While LE-SAM scalarized the gradient, the adversarial update is still on the gradient direction. The claim that LE-SAM "effectively removes the gradient interference" is too strong.
4. Some improvements are marginal compared to other baselines.
5. The scope of the experiments conducted is mainly on clean in-distribution tasks, which may not fully demonstrate the model's ability to generalize under distribution shifts. The author could consider evaluating the model on OOD or corrupted datasets to better showcase the model's robustness.
6. In between line 174 and 175 there seems to be a typo.

---

> ### Author Rebuttal · Authors · 2026-03-31
>
> Extra experimental results:https://anonymous.4open.science/r/rebuttalsam-F7B0.The figures and tables referred to in the rebuttal correspond to those in the repository
>
> W1:We have revised Figure 2 in main paper shown in Fig.6
>
> W2,W3:We agree that there are two different notions of sharpness as the reviewer mentioned:(i)The flatness of the attained minima at the end of the optimization process, which is inherently a second-order property.(ii)The information used during optimization process to reach such minima,where gradients still play an important role.We do not intend to claim that gradient information is unnecessary in optimization.In fact, similar to standard SAM,LE-SAM still constructs the perturbation along direction of center weight.Our point is more specific:under the standard stop-gradient treatment of the inner perturbation,LE-SAM scalarizes the first-order loss variation into a constant budget, so the gradient-norm term no longer dominates the outer surrogate, whereas in SAM the surrogate contains the explicit term $\rho\|\nabla L(w)\|_2$,which often dominates most of training process.Therefore,our claim is not that LE-SAM eliminates gradients,but it reduces the dominance of the gradient-norm-driven signal and makes the effective optimization more curvature-based.we further evaluate the ratio between first-order and second order term (shown in Fig.4) to better demonstrate this phenomenon that the second-order term contributes more significantly to the whole optimization process for LE-SAM
>
> W4:Unlike SAM variants,LE-SAM only redesigns the perturbation mechanism while keeping the overall framework unchanged. Therefore,achieving consistent improvements over SAM and its variants without adding extra complexity or components demonstrates that the proposed mechanism is both principled and practically effective;Meanwhile,Motivated by the reviewers' concern on annealing and clipping, we turn off the clipping and annealing and conduct the ablation.The results are shown in Tab.4.We found when we use relatively smaller $\sigma$ to ensure the convergence of the training,on WRN($\sigma=0.3$) and PyramidNet($\sigma=0.3$) the model performance can be further improved (84.43 to 84.91, 85.72 to 85.91),we provide the loss/acc curve across training in Fig.5;Moreover,inspired by reviewer 1CdC Q2,we extend LESAM to LESAM+ which can further improve the performance
>
> LESAM+:From eq.13 in main paper,since $\sigma$ is a scaler term and has zero gradient,the difference between $L_S(\mathbf{w} + \dot{\epsilon_t} )$ and $L_S(\mathbf{w})$ can be seen as a proxy to the second-order curvature term:
> \begin{equation}\hat{H}(\mathbf{w})= L_S(\mathbf{w} + \dot{\epsilon_t} )- L_S(\mathbf{w})\approx\underbrace{\sigma}_{\text{scalar constant}}+\frac{1}{2}\\dot{\epsilon}_t^\top H(\mathbf{w})\\dot{\epsilon}_t \end{equation}Thus,we can extend the optimization objective to $L_S(\mathbf{w} + \dot{\epsilon_t} )+\alpha \hat{H}(\mathbf{w})$ to amplify the curvature awareness where $\alpha$ is a hyperparameter,we select $\alpha$ as 0.5 0.55 0.6 with corresponding $\sigma$ as 0.15,0.2,0.25 on three training backbones.We evaluate the LESAM+ performance on CIFAR-100,the experiment result is shown in Tab.3,where the performance can be further improved(+0.68, +0.63, +0.47 on three training backbones). we also provide the loss/acc curve across training in Fig.2
>
> W5:We evaluate single domain generalization (SDG) performance and robustness under adversarial attack of LE-SAM.For SDG, we evaluate the performance on PACS and VLCS with a improved baseline for domain generalization ERM++ (Teterwak, 2025).The results are shown in Tab.1,LESAM can consistently surpass SAM an average of 2.18 and 0.84 on two datasets,showing better generalization performance. Under advesarial attacks, we follow (zhang, 2024) for CIFAR with $L_2$ and $L_{\infty}$ PGD attacks and compared with SAM.For SAM, we adopt suggested $\rho=0.4$,for LE-SAM,we choose 0.2,0.4 for $\sigma$ on CIFAR-10 and CIFAR-100.Since the paper and its workshop version did not mentioned the attack step size selection,we follow the hyperparameter selection in the eval.py in the paper's repository.The results are shown in Tab.2,LE-SAM can outperform 2.44,0.79,2.47 and 0.90 under all scenarios,showing stronger robustness under adversarial attacks, especially facing stronger attacks
>
> W6:We will fix in revised version
> Q1:We thank the reviewer for suggesting us using the SAM fallback.We wonder if the suggestion means using SAM only on the steps where the dynamic radius exceeds $\rho_{max}$,this is already effectively what radius clipping does in our implementation, since those clipped steps reduce to a fixed-radius SAM-style update.The only real difference would be a permanent handoff to SAM,which would constitute a different hybrid algorithm. Instead,when disabling both clipping and annealing,ablation experiment mentioned in W.4 show LE-SAM can still achieve comparable,and in some cases even achieve performance

---

> > ### Author Rebuttal · Reviewer_R7nK · 2026-03-31
> >
> > I'd like to remind the authors that the provided anonymous link does not show any content.
> >
> > The response resolves most of my concerns. For Q1, the concern is specifically about the annealing schedule, not the clipping. Clipping and annealing solve different problems. Clipping prevents the radius from exploding, while annealing turns off sharpness pressure entirely near convergence. The SAM fallback we proposed would replace the annealing to maintain sharpness pressure in a stable form rather than eliminating it. This is, to some extent, addressed by W4.

---

> > > ### Author Response · Authors · 2026-03-31
> > >
> > > We apologize for the confusion. Due to an automatic system parsing issue, the link in our rebuttal was incorrectly interpreted. Specifically, the system failed to properly separate:
> > >
> > > “Extra experimental results:”
> > >
> > >  https://anonymous.4open.science/r/rebuttalsam-F7B0
> > >
> > > “and”
> > >
> > > and instead merged them into a single unrecognized string.
> > >
> > > We would like to clarify that the correct anonymous repository is:
> > > https://anonymous.4open.science/r/rebuttalsam-F7B0
> > >
> > > The repository contains all figures and tables referenced in the rebuttal. Sorry for the confusion that might caused.
> > >
> > > Following the suggestion, we compare LESAM and LESAM (with SAM fallback) on ResNet-18, the results are shown in Fig.7(81.13 vs 81.15). Based on this observation, we tend to favor a self-contained formulation that avoids introducing additional mechanisms (such as fallback strategies or schedules). Thank you for the insightful suggestion.
> > >
> > > ——————Overall response: Thanks the Reviewers for their time and constructive feedback.————————
> > >
> > > We sincerely thank the reviewer R7nK for the constructive and insightful feedback. The main concerns raised focused on (1) the interpretation of “curvature-dominated” learning (2) the role of annealing and clipping (3) the limited evaluation scope, especially regarding robustness and out-of-distribution generalization.
> > >
> > > For(1), we clarified that our method does not eliminate gradient information used for model update and create the adversarial perturbation but instead reduces the dominance of the gradient-norm term in the optimization surrogate. We further provided empirical analysis comparing first and second-order contributions which trace the ratio between these two terms to support this interpretation.
> > >
> > > For(2), To address concerns about annealing and clipping, we conducted ablations without these components and showed that the core mechanism remains effective, meanwhile we found using smaller value of $\sigma$ can also ensure the convergence during training and can achieve better performance on Wideresnet and PyramidNet.
> > >
> > > For(3),  Finally, we expanded the experimental evaluation to include single domain generalization and robustness under adversarial attacks, which consistently support the effectiveness of our approach.
> > >
> > >  We thank the reviewer R7nK for pointing out the mistakes of our anonymous link due to an automatic system parsing issue and after we provide the correct link the reviewer increase our score and change the rebuttal acknowledgement to"fully resolved ", **sincerely wish you all the best in your submissions and future research.**

---

### Official Review · Reviewer_1CdC · 2026-03-13

**Soundness:** 3
**Presentation:** 3
**Significance:** 3
**Originality:** 3
**Overall Recommendation:** 4
**Confidence:** 4

**Summary:**

This paper studies the design of adversarial perturbations in Sharpness-Aware Minimization (SAM). The authors argue that standard SAM, which chooses a perturbation inside a fixed-radius parameter-space ball using a first-order approximation, produces a surrogate whose extra term is dominated by the gradient norm, whereas the notion of flat minima is more naturally tied to second-order curvature. To address this, the paper proposes Loss-Equated SAM (LE-SAM), which fixes a loss-space budget rather than a perturbation radius and then back-solves the corresponding perturbation magnitude in parameter space. The resulting perturbation radius becomes adaptive, shrinking or growing according to the current gradient norm, with additional clipping and late-stage annealing for stability. The paper motivates this reformulation through Taylor expansions and a Polyak-step-size analogy, and gives a simple training algorithm with the same two-pass structure as SAM. Empirically, LE-SAM is evaluated on CIFAR-10/100 with several architectures, on ImageNet with ResNet-50/101, on ViT-B-16 fine-tuning, and on CIFAR-100-LT in the appendix. The reported results show consistent improvements over SAM and several SAM variants, and the paper also includes Hessian-spectrum measurements, loss-landscape visualizations, perturbation-radius dynamics, and a late-stage-switch experiment intended to support the proposed mechanism.

**Compliance With Llm Reviewing Policy:**

Affirmed.

**Final Justification:**

The authors have managed to address my concerns. I will keep my positive score, and I will increase my confidence.

**Key Questions For Authors:**

1. The core interpretation of LE-SAM is that the first-order contribution becomes a fixed scalar budget, shifting the sharpness-aware signal toward curvature terms. How should this claim be interpreted once the practical stabilizers are active, especially when $\rho_t$ is clipped at $\rho_{max}$ as in Figure 5? This matters because the paper’s main conceptual argument appears exact only in the unclipped idealized surrogate.

2. The paper motivates LE-SAM as more curvature aware, but the perturbation direction remains gradient aligned. Can you clarify more precisely what notion of “curvature-dominated” learning you intend here, and whether you can empirically decompose the update into vanilla-gradient and sharpness-aware components?

**Limitations:**

Yes

**Strengths And Weaknesses:**

**Strengths**

1. The paper’s core method is clearly specified in Section 4 and Algorithm 1, and the main derivation is easy to follow. The transition from the standard SAM surrogate in Equation (6), where the additional term includes $\rho\|\nabla L_S(w)\|$, to the LE-SAM surrogate in Equation (13), where the first-order contribution is replaced by a fixed scalar budget $\sigma$, is intuitive and gives a coherent rationale for the method. Empirically, the paper evaluates LE-SAM on multiple datasets and model families, not just one benchmark.

2. The paper is generally readable and the conceptual contrast between “fix the radius” and “fix the loss” is communicated effectively, especially through Figures 1 and 2. The method is simple enough that the reader can understand the proposal without excessive notation, and the algorithm box plus hyperparameter tables in the appendix improve reproducibility.

3. Finally, the main contribution is not a small tweak to a SAM hyperparameter but a genuine reformulation of the perturbation rule. The Polyak-inspired loss-budget view is a fresh perspective relative to the standard fixed-radius interpretation, and the paper does a good job of connecting that change to a specific mechanistic claim about what part of the surrogate dominates learning.

**Weaknesses**

1. The main soundness weakness is that the paper’s central conceptual claim is stronger than the formal support currently provided. The “objective mismatch” argument in Section 3 and the “curvature-dominated” interpretation in Section 4 rely on local Taylor expansions and a stop-gradient convention, but the paper provides no theorem showing improved convergence, sharper flatness control, or better generalization under the proposed perturbation. In particular, the key statement that the first-order contribution becomes a constant is exact only for the idealized surrogate in Equation (13); in practice, the method introduces both a numerical stabilizer $\varrho$ and radius clipping
$\rho_{max}$, and Figure 5 indicates that clipping is active for a nontrivial part of training. In those regimes, the perturbation no longer exactly satisfies the intended fixed-budget relation, so the paper’s mechanism-level interpretation becomes approximate. This does not invalidate the method, but it weakens the force of the paper’s explanatory claims.

2. There are a few grammatical issues and typos, and some claims are stated too categorically relative to the evidence. For example, the paper sometimes suggests that LE-SAM “completely abandons” first-order information, but the perturbation direction is still gradient aligned; what is removed is the gradient-norm dependence of the extra first-order term in the local surrogate, not the role of gradients altogether. A more careful phrasing would improve precision.

3. I think the contributions are meaningful but not yet fully convincing as a definitive advance. The method is simple and potentially useful, but without stronger ablations and a more complete competitor set, it is hard to know whether this will substantially reshape how the community thinks about sharpness-aware training, or whether it is a strong but still somewhat specialized refinement. The impact is therefore positive but not yet clearly high.

---

> ### Author Rebuttal · Authors · 2026-03-31
>
> Our supplementary experimental results are available in the anonymous repository:https://anonymous.4open.science/r/rebuttalsam-F7B0. The figures and tables referred to in the rebuttal correspond to those in the repository.
>
>  W1 and Q1:The main motivation of clipping and annealing is to ensure training convergence of LE-SAM. To directly address this concern, we further conducted an ablation experiment without clipping and annealing.In this setting, we use a relatively smaller $\sigma$ to maintain stable convergence of training.The ablation results are shown in Fig.5 and Tab.4. We find that on ResNet-18 when we choose $\sigma=0.35$ the model can achieve comparable performance than with clipping and annealing, and on the other two architectures it can even achieve slightly better performance (we choose $\sigma=0.3$).This suggests that the core loss-equated perturbation itself remains effective.We will include this ablation and clarify that clipping/annealing are practical convergence safeguards, not the source of the main improvement.
>
> W2:We agree our current explanation might cause misunderstanding,thus we want to clarify two different notions of first-order information in our discussion: (i)What the reviewer mentioned is the gradient itself, which is still used to define the perturbation direction and the update direction in LE-SAM, just as in SAM which still plays an important role in whole optimization process. (ii)The second is the explicit first-order gradient-norm term arising in the local Taylor surrogate ($\rho\|L(\textbf{w})\|$)
>  term in SAM.Our intended claim is only that LE-SAM removes the dominance of this gradient-norm-dependent first-order term in the surrogate by scalarizing it into a constant budget under the stop-gradient treatment, not that it eliminates the role of both two notions of gradients altogether.For the typo and grammar mistake,we will fix them in the revised version.
>
> W3:For the ablation study,please see the rebuttal of W1 and Q1.Beyond current experiment,we evaluate single domain generalization (SDG) performance (using one domain as the source domain and generalize on the rest domains) and  robustness under adversarial attack of LE-SAM.For SDG, we evaluate the performance on PACS (pretrained ResNet-18) and VLCS (ResNet-18 without pretraining) with a improved baseline for domain generalization ERM++ (Teterwak, 2025). The results are shown in Tab.1, LESAM can consistently surpass SAM an average of 2.18 and 0.84 points on two datasets, demonstrating better generalization performance.For robustness under advesarial attacks, we follow (zhang, 2024) for CIFAR 10/100 with $L_2$ and $L_{\infty}$ PGD attacks and compared with SAM. For SAM, we adopt suggested $\rho=0.4$ following (Zhang, 2024), for LE-SAM, we choose 0.2, 0.4 for $\sigma$ on CIFAR-10 and CIFAR-100.Since the paper (zhang, 2024) and its workshop version did not mentioned the attack step size selection on the test set, we follow the hyperparameter selection in the eval.py in the paper's repository.The results are shown in Tab.2,LE-SAM can outperform 2.44 0.79 2.47 and 0.90 points under all scenarios, showing stronger robustness under adversarial attacks,especially facing stronger attacks.
>
> Q2:We agree that our current explanation might cause misleading.As also clarified in our response to W2,our discussion involves two different notions of first-order information:(i) the gradient itself,which still determines the perturbation and update directions in LE-SAM, and (ii) the explicit gradient-norm term in the local Taylor surrogate term in SAM.Our intended claim is only that LE-SAM removes the dominance of the latter term in the surrogate by scalarizing it into a constant budget under the stop-gradient treatment, not that it removes gradient information altogether.We will revise the wording in the revised version.Moreover, inspired by the suggestion of decomposing the gradients in Q2,we proposed LESAM+ which described in the following paragraph.
>
> LESAM+: From eq.13 in main paper, since $\sigma$ is a scaler term and has zero gradient,the difference between $L_S(\mathbf{w} + \dot{\epsilon_t} )$ and $L_S(\mathbf{w})$ can be seen as a proxy to the second-order curvature term:
> \begin{equation}
>  \hat{H}(\mathbf{w})= L_S(\mathbf{w} + \dot{\epsilon_t} )- L_S(\mathbf{w})
> \approx
> \underbrace{\sigma}_{\text{scalar constant}}+
> \frac{1}{2}\ \dot{\epsilon}_t^\top H(\mathbf{w})\ \dot{\epsilon}_t .
> \end{equation}
> Thus,we can extend the optimization objective to $L_S(\mathbf{w} + \dot{\epsilon_t} ) + \alpha \hat{H}(\mathbf{w})$ to amplify the curvature awareness where $\alpha$ is a hyperparameter,we select $\alpha$ as 0.5 0.55 0.6 with corresponding $\sigma$ as 0.15 0.2 0.25 on three training backbones.We evaluate the LESAM+ performance on CIFAR-100, the experiment result is shown in Tab.3, where the performance can be further improved(+0.68, +0.63, +0.47 on three training backbones).we also provide the loss/acc curve across training in Fig.2.

---

> > ### Author Rebuttal · Reviewer_1CdC · 2026-04-01
> >
> > Thank you for your detailed response. The authors have managed to address my concerns. I will keep my positive score and I will increase my confidence.

---

> > > ### Author Response · Authors · 2026-04-02
> > >
> > > Thank you very much for your recognition of our work and valuable suggestions！
> > >
> > >
> > > ——————Overall response: Thanks the Reviewers for their time and constructive feedback.————————
> > >
> > > We sincerely thank the reviewer 1CdC for their time and thoughtful feedback.
> > > The main concerns focused on: (1) whether our “curvature-dominated” interpretation; (2) whether the mechanism still holds under practical stabilizers such as clipping and annealing;  (3) the precision of our claims regarding the role of first-order information.
> > >
> > > For (2), we provided additional ablation results removing clipping and annealing to demonstrate that the core loss-equated perturbation remains effective, clarifying that the clipping and annealing is only for the stability of training and can be realized through smaller $\sigma$ which even can achieve better performance on Wideresnet and PyramidNet.
> > >
> > > For (1) and (3) , We  refine our explanation by distinguishing between gradient direction and the gradient-norm term in the surrogate, emphasizing that our method removes the dominance of the latter rather than eliminating gradients altogether.
> > >
> > > Beyond the concerns that reviewer mentioned, we strengthened the empirical evidence with additional evaluations (e.g., robustness and domain generalization) and introduced LESAM+ to further support our mechanism.
> > >
> > >  Overall, we believe our respones address the reviewer’s concerns and clarify both the theoretical interpretation and practical effectiveness of our approach. We thank the reviewer 1CdC again for the valuable suggestions, **sincerely wish you all the best in your submissions and future research.**

---

### Official Review · Reviewer_nCae · 2026-03-13

**Soundness:** 3
**Presentation:** 3
**Significance:** 3
**Originality:** 3
**Overall Recommendation:** 4
**Confidence:** 4

**Summary:**

This paper argues that, in SAM, the ascent direction can sometimes induce excessively large losses, which may lead to undesirable training behavior. The authors provide a mathematical analysis of this phenomenon and propose LE-SAM, a loss-aware \rho scheduling strategy designed to address it. The experimental setup and results are cleanly presented without unnecessary complexity. However, I recommend including comparisons with several related approaches that also modify the \rho schedule.

**Compliance With Llm Reviewing Policy:**

Affirmed.

**Final Justification:**

Thank you for the authors’ detailed and thoughtful responses. The clarifications provided have satisfactorily addressed all of my concerns. However, I am not fully convinced that the level of novelty warrants a strong acceptance. While I appreciate these improvements and clarifications, I will maintain my original score. I thank the authors for their efforts and their thorough rebuttal.

**Key Questions For Authors:**

Refer to Strengths And Weaknesses

**Limitations:**

The authors did not discuss the limitations of their proposed method. I encourage the authors to include a discussion on potential limitations.

**Strengths And Weaknesses:**

### Soundness

Strength.

- The issue illustrated in Figure 1 clearly highlights the core motivation of the paper. This explanation connects well with the final paragraph of the Related Work section and helps the reader understand the central claim of the paper. This motivation is also well aligned with the development in Section 3.

Weakness.

- In Line 144, the authors note that “Most of the training process is dominated by the first-order gradient norm.” It is not entirely clear why this behavior occurs, and additional explanation would help readers better understand this claim. Providing empirical plots comparing the magnitude of the first-order gradient term and the Hessian-related term during training could further clarify this point.

### Presentation

Strength.

- The overall presentation is very clean and concise. The paper is easy to read and well structured, and the flow of the arguments is particularly clear. The experimental results are also well visualized.

Weakness.

- Figure 2 is difficult to read in the current format. Including it as a PDF figure may improve the visual quality.
- In Table 1 and Table 2, it would be helpful to highlight the best results in bold for easier comparison.

### Significance

Strength.

- Connecting the Loss-Equated Perturbation idea with the Polyak step-size is a particularly interesting contribution. To the best of my knowledge, this perspective has not been explored in this way before, which makes the approach conceptually appealing.

Weakness.

- I am curious about the computational overhead. Since the proposed method appears to introduce additional computations compared to standard SAM, it would be helpful to report training time or computational cost relative to existing methods.

### Originality

Strength.

- N/A

Weakness.

- The paper emphasizes fixing the loss rather than the radius. However, when examining Algorithm 1 carefully, the resulting procedure effectively behaves similarly to \rho annealing. Several prior works have also attempted to improve SAM by modifying the \rho schedule. Therefore, comparisons with related approaches would strengthen the paper. It would also be helpful to show what happens when this budget annealing mechanism is removed. In particular, comparisons with the following works may be useful:
    - Zou, Jinping, Xiaoge Deng, and Tao Sun. "Sharpness-aware minimization with adaptive regularization for training deep neural networks." ICASSP 2025-2025 IEEE International Conference on Acoustics, Speech and Signal Processing (ICASSP). IEEE, 2025.
    - Tan, Chengli, et al. "Stabilizing sharpness-aware minimization through a simple renormalization strategy." Journal of Machine Learning Research 26.68 (2025): 1-35.

Question.

- I also wonder whether this clipping behavior could be related to the saddle-point problem in SAM. Recent studies (When do flat minima optimizers work? [NeurIPS 2022], Stability analysis of sharpness-aware minimization [arXiv]) suggest that the SAM radius can be connected to saddle-point behavior. It would be interesting to investigate whether \rho clipping could alleviate this issue. If such a connection exists, it could potentially lead to an important experimental or theoretical insight.

---

> ### Author Rebuttal · Authors · 2026-03-31
>
> Our extra experimental results: :https://anonymous.4open.science/r/rebuttalsam-F7B0. The figures and tables referred to in the rebuttal correspond to those in the repository
>
> Weakness of Soundness:The the perturbed loss expansion contains both a first-order term and a second order term $\frac{1}{2}\ \hat{\epsilon}^\top H(\textbf{w})\hat{\epsilon}$, during much of the early-to-middle training phase, when the gradient norm is still relatively large, the overall scale of the perturbed loss increase is typically governed primarily by the first-order gradient-magnitude term, whereas the curvature term becomes relatively comparable only later in training as gradients become smaller, this matches the recent observation by (Zhou, 2024) that SAM efficiently selects flatter minima late in training.Follow the reviewer's suggestion, we provide empirical plots comparing the ratio of the first-order gradient term and the second order term during training ($\frac{ \rho \|\nabla L_\mathcal{S}(\textbf{w}) \|_2
> }{ \frac{1}{2}\,\hat{\epsilon}^\top H(\textbf{w})\hat{\epsilon}}$) for SAM and LESAM in Fig.4.As shown in Fig.4 We observe that for SAM, the first-order term consistently dominates the second-order term throughout most of training,with an average ratio of 2.30.In contrast, for LE-SAM, the ratio is reduced to 1.36,which confirms that the second-order term contributes more significantly to the whole optimization process.
>
> Weakness of presentation:We will follow the reviewer's suggestion and fix all the problems in the revised version.we provide revised Figure 2 in main paper shown in Fig.6 in our repository
>
> Weakness of Significance:The additional overhead for LE-SAM lies in the calculation of the dynamic radius $\rho_t = \sigma/(\|\mathbf{g}\|_2 + \varrho)$.This involves a vector norm computation and a few scalar operations.We have measured both training time and memory footprint relative to SAM,and the results are summarized in Tab.5.LE-SAM has nearly identical memory usage to SAM and only introduces lightweight scalar operations for computing $\rho_t$,resulting in a negligible time overhead of about 1%--3%
>
> Weakness of Originality:We thank the reviewer for this helpful suggestion.However, even with annealing, the radius in LE-SAM is not heuristically scheduled,but is always back-solved from the loss budget $\sigma$ which is a consequence of the loss-equated mechanism rather than a pure radius schedule strategy.In comparison,SAMAR mainly adapts the regularization weight,and SSAM improves stability through gradient renormalization; neither changes the basic fixed-radius perturbation mechanism for SAM.By contrast, our method modifies the perturbation construction mechanism itself.We will clarify this in the revised version.Moreover, we conduct ablation study that turn off all the clipping and annealing,the results are shown in Tab.4 We found when we use relatively smaller $\sigma$ to ensure the convergence of the training, on WRN ($\sigma=0.3$) and PyramidNet($\sigma=0.3$) the model performance can be further improved (84.43 to 84.91, 85.72 to 85.91),we provide the loss/acc curve across training in Fig.5;Moreover,inspired by reviewer 1CdC Question2,we extend LESAM to LESAM+ which can further improve the generalization performance
>
> LESAM+:From eq.13 in main paper,since $\sigma$ is a scaler term and has zero gradient,the difference between $L_S(\mathbf{w} + \dot{\epsilon_t} )$ and $L_S(\mathbf{w})$ can be seen as a proxy to the second-order curvature term:
> \begin{equation}
>  \hat{H}(\mathbf{w})= L_S(\mathbf{w} + \dot{\epsilon_t} )- L_S(\mathbf{w})\approx\underbrace{\sigma}_{\text{scalar constant}}+
> \frac{1}{2}\ \dot{\epsilon}_t^\top H(\mathbf{w})\ \dot{\epsilon}_t
> \end{equation}Thus,we can extend the optimization objective to $L_S(\mathbf{w} + \dot{\epsilon_t} ) + \alpha \hat{H}(\mathbf{w})$ to amplify the curvature awareness where $\alpha$ is a hyperparameter, we select $\alpha$ as 0.5 0.55 0.6 with corresponding $\sigma$ as 0.15 0.2 0.25 on three training backbones.We evaluate the LESAM+ performance on CIFAR-100, the experiment result is shown in Tab.3, where the performance can be further improved(+0.68, +0.63, +0.47 on three training backbones).we also provide the loss/acc curve across training in Fig.2
>
> Q1:Thank you for this insightful suggestion.We agree that this is a very interesting connection.Recent analysis indeed suggest that SAM’s radius is closely related to saddle-point behavior and escape dynamics.This makes it plausible that our $\rho$-clipping/annealing may help alleviate such behavior.In addition,when we turn off the clipping and annealing or for LE-SAM+,we are tend to use relatively smaller $\sigma$,thus we agree there might exists a very interesting connection.However, the main motivation for us to adopt such clipping/annealing strategy is to ensure the training convergence,we think more detailed theoretical analysis can be discovered in the future work which we think will be very interesting.

---

> > ### Author Rebuttal · Reviewer_nCae · 2026-04-03
> >
> > Thank you for the authors’ detailed and thoughtful responses. The clarifications provided have satisfactorily addressed all of my concerns. However, I am not fully convinced that the level of novelty warrants a strong acceptance. While I appreciate these improvements and clarifications, I will maintain my original score. I thank the authors for their efforts and their thorough rebuttal.

---

> > > ### Author Response · Authors · 2026-04-03
> > >
> > > Thank you very much for your recognition of our work and valuable suggestions！
> > >
> > >
> > >
> > > ——————Overall response: Thanks the Reviewers for their time and constructive feedback.————————
> > >
> > > We sincerely thank the reviewer nCae for the constructive and insightful feedback. In the rebuttal, we have carefully addressed all the major concerns including Soundness, Significance, Originality, Presentation.
> > >
> > >  For **Soundness**, ( the soundness of our claim regarding first-order dominance), we provided both theoretical clarification and new empirical evidence which analyze the ratio of first-order and second-order terms to better justify our claims.
> > >
> > >  For **Significance**(the computational cost), we reported additional results on computational overhead, showing that our method introduces only negligible cost.
> > >
> > >  For **Originality**, (the novelty compared to existing SAM-based ρ-scheduling methods), we clarified that our approach is not simply ρ-scheduling strategy but a mechanism-level reformulation that back-solves the perturbation from a fixed loss budget inspire by Polyak optimal stepsize, and we further supported this with additional ablation studies (removing scheduling stategies: clipping/annealing)  and the extended LESAM+ variant.
> > >
> > >  For **Presentation** , we revised figures and tables for improved clarity.
> > >
> > > We believe the rebuttal has addressed the reviewer’s concerns and strengthened both the empirical and conceptual aspects of the work. Regarding novelty, while we acknowledge that our method shares connections with prior SAM variants, we emphasize that our contribution lies in a principled but simple, mechanism-level redesign of the perturbation objective, which we believe offers a distinct and meaningful perspective for the field of Sharpness-Aware Minimization (SAM).
> > >
> > > Overall, we thank the reviewer nCae for the positive recognition of our work and valuable suggestions, **sincerely wish you all the best in your submissions and future research.**

---

### Decision · Program_Chairs · 2026-04-30

**Decision:**

Accept (regular)

**Comment:**

The paper proposes Loss-Equated SAM (LE-SAM), which inverts the traditional SAM mechanism that fixed perturbation radius with a fixed loss-space budget, effectively removing gradient-norm–dominated learning signals and shifting optimization toward curvature-dominated terms. The paper also presents extensive experiments demonstrating LESAM's strong generalization ability.

All reviewers find the results of this work interesting and are in favor of the paper's acceptance. They give scores of 4, 4, 5, and 5. The authors provided more details during the rebuttal and discussion period, and all reviewers agreed that their concerns were fully resolved.

For the camera-ready, the AC advises the authors to incorporate the feedback received during the review process into the revised manuscript. The paper is well written and has clear contributions that could be of interest to the general optimization and ML communities. I suggest acceptance of this work.